# The pattern of long-term changes in the moisture transport for precipitation with Arctic sea ice melting

Luis Gimeno-Sotelo[1], Raquel Nieto[2], Marta Vázquez[2], Luis Gimeno[2]

[1]Facultade de Matemáticas, Universidade de Santiago de Compostela, 15782 Spain.
[2]Environmental Physics Laboratory (EphysLab), Universidade de Vigo, Ourense, 32004, Spain

*Correspondence to*: Luis Gimeno (l.gimeno@uvigo.es)

**Abstract**. In this study we use the term moisture transport for precipitation (MTP) for a target region as the moisture coming to this region from its major moisture sources that then results in precipitation over it. We have identified the patterns of change in moisture transport for precipitation over the Arctic region, the Arctic Ocean, and its 13 main subdomains concurrent with

the major sea ice decline occurred in 2003. The pattern consists of a general decrease in moisture transport in summer and enhanced moisture transport in autumn and early winter, with different contributions depending on the moisture source and ocean subregion. The pattern is not only statistically significant but also consistent with Eulerian fluxes diagnosis, changes in the frequency of circulation types, and any of the known mechanisms of the effects of the increments of precipitation as snowfall or rainfall on ice in the Arctic. The results of this paper also reveal that the assumed and partially documented

enhanced poleward moisture transport from lower latitudes as a consequence of increased moisture from climate change seems to be less simple and constant than typically recognized in relation to enhanced Arctic precipitation throughout the year in the present climate.

## 1 Introduction

The shrinking of the cryosphere since the 1970s is among the most robust signals of climate change identified in the last IPCC Assessment *(IPCC, 2013)*. There is little doubt that this change is a result of global warming caused by increased anthropogenic greenhouse gas emissions *(AR5; IPCC, 2013)*. The Arctic is of particular scientific and environmental interest; the rise in Arctic near-surface temperature doubles the global average in almost all months *(e.g., Screen and Simmonds, 2010; Tang et al., 2014, Cohen et al., 2014)*. Without doubt, the most important indicator of Arctic climate change is sea ice extent, which is

characterised by a very significant decline since the 1970s *(Tang et al., 2014; IPCC, 2013)*. This decline has accelerated in recent decades in terms of both extent and thickness to the point where a summer ice-free Arctic Ocean is expected to occur within the next few decades *(IPCC, 2013)*. Changes in atmospheric and oceanic circulation with implications for Arctic mid-latitude climate or reductions and shifts in the distribution of oceanic and terrestrial fauna are among the most concerning and already apparent impacts of this decline in sea ice (Screen and Simmonds, 2010; Post et al, 2013). The scientific mechanisms

involved in Arctic sea ice extension (SIE) are multiple and varied; atmospheric processes *(Ogi and Wallace, 2007; Rigor et al., 2002)* interact closely and nonlinearly with hydrological and oceanographic processes *(Zhang et al.,1999; Årthun et al., 2012; Zhang et al., 2012)*. Changes in atmospheric circulation can affect SIE via dynamical (e.g. changes of surface winds) or thermodynamic factors (i.e. changes in heat and moisture fluxes in the Arctic *(Tjernström et al., 2015; Ding et al., 2017)*.

*Am*ong these effects, change in moisture transport has emerged as one of the most important with respect to the greenhouse effect *(Koenigk et al., 2013; Graversen and Burtu, 2016; Vihma, 2016)*, and is related to SIE decline through hydrological mechanisms such as changes in Arctic river discharges *(Zhang et al., 2012)*, radiative mechanisms such as anomalous downward longwave radiation at the surface *(Woods et al., 2013; Park et al., 2015a; Mortin et al., 2016; Woods and Caballero, 2016; Lee et al., 2017)*, or meteorological mechanisms such as changes in the frequency and intensity of cyclones crossing the

Arctic *(Rinke et al., 2017)*. Because the effects of enhanced moisture transport on Arctic ice are diverse, there is no direct relationship between enhanced transport and SIE decline. Anomalously high moisture transport into the Arctic is associated with intense surface winds, and increment in moisture content and induced radiative warming, which lead to decreased SIE *(Kapsch et al., 2013; Park et al., 2015b)*. However, anomalous moisture transport can result in anomalous precipitation, and in this case, the relation between enhanced moisture transport and diminished SIE is unclear because changes in precipitation

are not always related to SIE in the same way, depending on the type of precipitation and the season. Snowfall on sea ice enhances thermal insulation and thus reduces sea ice growth in winter *(Leppäranta, 1993)*, but increases the surface albedo and thus reduces melt in spring and summer *(Cheng et al., 2008)*. In contrast, rainfall is generally related to sea ice melt, and for both snowfall and rainfall, flooding over the ice favours the formation of superimposed ice and potentially increases in the Arctic sea ice thickness.

In this study we use the term moisture transport for precipitation (MTP) for a target region as the moisture coming to this region from its major moisture sources that then results in precipitation over it. We focus on identifying the patterns of moisture transport for precipitation in the Arctic region (AR), the Arctic Ocean (AO) as a whole, and its 13 main subdomains, which fit better with sea ice decline. For this purpose, we have studied the different patterns of moisture transport for the cases of high and low SIE linked to periods before and after the main change point (CP) in the extension of sea ice. The study differs

significantly from our previous work, *Vazquez et al. (2017),* in which we analyzed the influence of the transport of moisture on the two most important sea ice minimum events (2007 and 2012) based mostly on an analysis of anomalies. In this paper we analyzed the long-term changes in the moisture transport concurrent with long-term changes in sea ice (sea ice decline).

## 2 Data and Methods

### 2.1 Data

The target region in this study comprises the AR (figure 1a) as defined by Roberts et al. (2010) and used by *Vazquez et al.*
*(2016)*, the AO as a whole, and its 13 main subdomains as defined in figure 1b (*Boisvert et al., 2015*). The analysed period

was from 1 January 1980 to 31 December 2016. For Arctic SIE, we used daily, monthly and annual data from the U.S. National

Snow and Ice Data Center (*Fetterer, 2016*), and to implement the Lagrangian approach, by which moisture transport is

estimated to approximate vertical integrated moisture fluxes and circulation types, we used the European Centre for Medium-

Range Weather Forecasts (ECMWF) interim Re-Analysis (ERA-Interim) *(Dee et al., 2011)*. Data from this reanalysis cover
the period from January 1979 to the present and extending forward continuously in near-real time. These data are available at

six-hour intervals at a 1° x 1° spatial resolution in latitude and longitude for 61 vertical levels (1000 to 0.1 hPa). The ERA-

Interim reanalysis is typically considered to have the highest quality relative to other reanalysis data for the water cycle *(Lorenz*

*and Kunstmann, 2012)* and is especially appropriate over the Arctic region *(Jakobson et al., 2012),* with better representation

of mass fluxes including water vapor *(Graversen et al., 2011).*

### 2.2 Methods

#### 2.2.1 Calculation of sea ice extension anomalies

Daily, monthly and annual data of sea ice extension from 1980 to 2016 taken from the National Snow and Ice Data Center

*(Fetterer, 2016)* were used to build four different series of ice extension anomalies for the whole Arctic Ocean (AO) and its
13 subregions. These four series were constructed as follows:

**DS**: 365 daily anomaly series (size of each series: 37 data points); for each daily value, the average for the same calendar day

over the 37-year period is subtracted (for instance, the series for 21 September comprises the anomaly of each 21 September

vs. the average of the 37 data points from this date)

**MS:** 12 monthly anomaly series (size of each series: 37 data points); for each monthly value, the average for the same month
over the 37-year period is subtracted (for instance the series for September results of the anomaly of each September vs the

average of the 37 September data points).

**ADS:** one series of all daily anomalies (size of the series: 13,505 data points) built by ordering the daily anomalies in DS from

1 January 1980 to 31 December 2016.

**AMS:** one series of all monthly anomalies (size of the series: 444 data points) built by ordering the monthly anomalies in MS
30  from January 1980 to December 2016.

## 2.2.2 Detection of change points in Arctic sea ice extension

We have used several methods to estimate change points in Arctic sea ice (ASI) extension to detect when the main long-range change occurred. As usual in time series analysis a change point detection tries to identify times when the time series in mean or variance changed. In this case we were interested mainly in changes in mean (in the Arctic sea ice extension the change in
mean is equivalent to a decrease, higher ASI extension values before the change point and lower after it).

Change points of each of these series for the AO were calculated using three different methods, one detecting single change points (At Most One Change, AMOC) and two detecting multiple change points (BinSeg and PELT).

AMOC uses a test to detect a hypothesised single change point (CP). The null hypothesis refers to no change point, and its maximum log-likelihood is given by $log\,p\,(y_{1:n}|\widehat{\theta_1})$, where p(·) is the probability density function associated with the
10 distribution of the data and $\widehat{\theta}$ is the maximum likelihood estimate of the parameters. For the alternative hypothesis, a change point at $\tau_1$ is considered, with $\tau_1 \in \{1, 2, ..., n-1\}$. The expression for the maximum log-likelihood for a given $\tau_1$ is $ML(\tau_1) = \log p(y_{1:\tau_1}|\widehat{\theta_1}) + \log p(y_{(\tau_1+1):n}|\widehat{\theta_2})$.

Taking into account that the CP location is discrete in nature, $\max \tau_1\, ML(\tau_1)$ is the maximum log-likelihood value under the alternative hypothesis, where the maximum is taken over all possible change-point locations. Consequently, the test statistic
is $\lambda = 2[\max \tau_1\, ML(\tau_1) - \log p(y_{1:n}|\widehat{\theta})]$. The null hypothesis is rejected if $\lambda > C$, where C is a threshold of our choice. When detecting a CP, its position is estimated as $\widehat{\tau_1}$, which is the value of $\widehat{\tau_1}$ that maximises $ML(\tau_1)$.

BinSeg and PELT: by summing the likelihood for each of the $m$ segments, the likelihood test statistic can be extended to multiple changes. However, there is a problem in identifying the maximum of $ML(\tau_{1:m})$ over all possible combinations of $\tau_{1:m}$. The most common approach to resolve this problem is to minimise
$\sum_{i=1}^{m+1}[C(y_{(\tau_{i-1}+1):\tau_i})] + \beta f(m)$ (1)

where C is a cost function for a segment, such as the negative log-likelihood, and $\beta f(m)$ is a penalty to guard against over fitting. The BinSeg (binary segmentation) method starts with applying a single CP test statistic to the entire data set. If a CP is identified, the data are then split into two at the CP location. The single CP process is repeated for the two new data sets, before and after the change. If CPs are identified in either of the new data sets, they are split further. This procedure continues until
no CPs are found in any parts of the data. This process is an approximate minimisation of (1) with $f(m) = m$, as any CP locations are conditional on CPs identified previously. The segment neighbourhood algorithm precisely minimises the expression given by (1) using a dynamic programming technique to obtain the optimal segmentation for $(m + 1)$ CPs reusing the information calculated for $m$ change points. The PELT (pruned exact linear time) method is similar to that of the segment neighbourhood algorithm in that it provides an exact segmentation. We have used the PELT algorithm instead of the segment
neighbourhood method as it has proven to be more computationally efficient. The reason for this greater efficiency is the use of dynamic programming and pruning in the algorithm's construction.

A full description of these methods and the subroutines in R used in this study can be found in *Killick and Eckley (2014)*.

The three different change point methods were used for the four different ASI ice extension time series defined in point 2.2.1. Figure 2 illustrates the detection of the change point in mean for two series (ADS and AMS) using AMOC method. The top plot represents the 13505-values Arctic ice extent anomalies series consisting of all days from 1st January 1980 to 31st December 2016 (ADS series). There are two horizontal lines representing the mean of the values before and after the change point identified by the AMOC method (the 8660th day that correspond to 22th September 2003). Those means are 0.27 and -0.91, respectively. The graphic at the bottom portrays the 444-values Arctic ice extent anomalies series consisting of all months from January 1980 to December 2016 (AMS series). As in the previous one, there are two horizontal lines, which correspond to the mean of the data before and after the AMOC change point (the 286th month of the series, that is -October 2003). Those means are 0.41 and -0.74, respectively.

### 2.2.3 Estimation of the Lagrangian moisture transport from the main sources

In this study, we used a Lagrangian approach to calculate moisture transport from the main moisture sources as detected by *Vazquez et al. (2016)* (figure 1c) for the AR, AO, and its 13 subregions. This approach is based on the particle dispersion model FLEXPART v9.0 (i.e., the FLEXible PARTicle dispersion model of Stohl and James (2004, 2005)) forced by ERA-Interim data from the ECMWF. This approach has been used extensively in moisture transport analysis *(e.g., Gimeno et al., 2010; 2013)*, and a review of its advantages and disadvantages versus other approaches for tracking water vapor was summarised by *Gimeno et al. (2012 and 2016)*. To briefly summarise this method, the atmosphere is divided into so-called particles (finite elements of volume with equal mass) and individual three-dimensional trajectories are tracked backward or forward in time for 10 days, the average residence time of water vapor in the atmosphere *(Numaguti, 1999)*. Then, taking into account the changes in ($q$) for each particle along its trajectory, the net rate of change of water vapour ($e - p$) for every particle, ($e - p$) = $m(dq/dt)$, is estimated, with $e$ and $p$ representing evaporation and precipitation, respectively. The total atmospheric moisture budget ($E - P$) is estimated by adding up ($e - p$) for all particles over a given area at each time step used in the analysis. If we follow the particles backward in time for a target region, positive (E-P) values identify the main moisture sources for this target region, if we follow the particles forward in time from a source region, negative (E-P) values identify the main moisture sinks of the source region. In this study we have used four predefined moisture sources, those identified as the major sources of moisture for the Arctic region (AR) in Vazquez et al (2016) following backward in time all the particles reaching the AR for the period 1980-2012 and taking the regions showing positive (E-P) values greater than 90th percentile (taking into account global values of positive E-P). Then, to compute moisture transport for precipitation (MTP) from each of these four sources to each sink for the AO, the trajectories of particles from the moisture sources for the Arctic (AR) were followed forward in time from every source region detected by *Vazquez et al. (2016)* (figure 1c)..This was done for 6 hours in the period 1980–2016. Then, we selected all particles losing moisture, (e − p) < 0, at the sinks (whole Arctic or any of the sub-regions), and by adding e − p for all of these particles, we estimated moisture transport for precipitation from the source to the sink ((E − P) < 0) at daily, monthly or yearly scales. A schematic illustration of this approach is displayed in

figure 3. A couple of clarifications on this approach are necessary to avoid misunderstandings of the results. The first one is that particles can gain moisture in the regions placed between the defined as the major moisture sources and the target region, even in the target region itself. However as our defined moisture regions were identified as the major moisture sources in the backward analysis (Vazquez et al, 2016) the contribution of the intermediate regions is much lower. A look at figure 2 from

Vazquez et al.(2016) shows that intermediate regions are not net sources (particles reaching the Arctic region lost (not gained) in average moisture in these regions. In any case not all the precipitated water comes from the major sources, those particles that were not within those major source regions ten days before precipitation are responsible for the rest of precipitation, including the particles coming from the own Arctic region, which account for the important contribution of local moisture re-circulation. The second clarification is concerned with the size of the target regions and the number of particles reaching them.

As commented in the seminal description of the approach *(Stohl and James, 2004, 2005),* this works better for large regions. The size of the target regions in this study (Arctic region and subregions) is bigger than in many of the regions where the same methodology was used in previous studies *(e.g. Ramos et al., 2016 or Wegmann et al, 2015)* and the average number of particles by source that reach daily the target regions big enough (see Table S1 in the supplementary material).

### 2.2.4 Identification of circulation types

The patterns of (E − P) < 0 can change daily in association with variations in atmospheric circulation. To evaluate the relationship between this variability and the moisture supply generated from each source of moisture, we identified different circulation types *(CTCs)* over the areas of interest using a methodology developed by *Fettweis et al. (2011).* This approach

consists of an automated circulation type classification based on a correlation analysis whereby atmospheric circulation is categorised into a convenient number of four discrete circulation types in the present study. For each pair of days, a similarity index based on the correlation coefficient was calculated for the purpose of grouping days that showed similar circulation patterns *(Belleflamme et al., 2012).* The first category contains the greatest number of similar days, where similarity is defined by a particular threshold (0.95 for the first class). After establishing the first class, the same procedure was applied for the

remaining days using a lower similarity threshold to find the second and then all other classes. The complete procedure was repeated for different thresholds to optimise the percentage of variance explained (Philipp et al., 2010). This method is termed a "leader" algorithm because each class is represented by a leader pattern considered as the reference day (Philipp et al., 2010). In this study, we used the geopotential height field at 850 hPa from ERA-Interim. Because our aim was to find and analyse different circulation patterns affecting the Arctic system linked to each source of moisture, the northern hemisphere from 20º

to 90ºN was divided into sections according to the positions of the major sources of moisture (figure 1c). The size of the sections was 70 ºlatitude x 90ºlongitude. Thus, the circulation type classification was obtained seasonally for each of these sections. The use of a regional domain centered in the moisture source is justified to account for regional modes instead of annular ones, which could not catch details in regional circulation. As changes in the size of the sections can vary lightly the

circulation types *(Huth et al., 2008)* we performed a sensibility analysis (not shown)  by  moving the domain 10ª eastward and westward and by extending the domain 10º eastward with (similar results, what showed that  the patterns are very coherent).

## 3 Results

### 3.1 Climatological moisture transport for precipitation (MTP) to the Arctic region (AR) and Arctic Ocean (AO) subregions

Figure 4 displays the seasonal cycle of MTP to the AR from four major sources (Atlantic, Pacific, Siberia, and North America) and the relative contributions of each. MTP to the AR exhibits a marked seasonal cycle with a maximum in summer and a

minimum in winter; MTP doubles in the highest month (August) relative to the lowest month (February). The percentage of the contribution from each source is relatively constant throughout the year, with the Pacific, North America, Siberia, and the Atlantic contributing about 35%, 30%, 20%, and 15%, respectively.

The seasonal cycle is quite similar for eight of the 13 AO subregions (figure 5), with two other regions (Barents and the Central Arctic) exhibiting similar maxima, but with minima in May, and another (Chukchi) with two minima in March and November,

and finally Greenland with a minimum in May but no clear maximum in summer, extending the typical summer high values to fall and winter. The relative contributions of each moisture source are very diverse, with three general patterns: one with the closest source dominating (a Pacific source for Bering, Chukchi, and Okhostk, a North American source for the Canadian Islands and Hudson, and a Siberian source for Kara and Laptev), another pattern where two sources share importance throughout the year (Atlantic and Siberian sources for Baffin and Greenland, and Pacific and North American sources for

Beaufort), and a third where the Siberian source shares importance with two or more sources (Barents, Central Arctic, and East Siberia).

The relative importance of each ocean subregion on MTP to the AO (estimated as the sum of the 13 subregion) is shown in figure 6 where the percentage of moisture transport to the whole AO is displayed by region and month. There are four regions where the aggregated MTP received represents more than 60% of the MTP received for the whole Arctic. Those regions are

Greenland, Baffin, Bering, and the Central Arctic, in order of importance. The contributions of Baffin, Bering, and the Central Arctic vary little throughout the year, representing about 20%, 12% and 10% of the MTP received for the whole Arctic, respectively, whereas the contribution of Greenland has a marked seasonal cycle with values around 25% for fall and winter, and 10–15% in spring and summer, seasons when two other sources gain importance, Hudson and Okhotsk, with percentages close to 10%. The almost constant contribution of Barents throughout the year is non-negligible, around 5%. The remaining

sources have contributions lesser than 5%.

### 3.2 Moisture transport after and before Arctic sea ice change points

Figure 7 summarises the identified change points (CPs) in means identified using the AMOC method in the four series of whole Arctic sea ice extent anomalies (DS, MS, ADS, and AMS) from 1980 to 2016.

Blue points refer to the change points in DS; for instance, the 21 July daily anomaly series (size of the series: 37 data points, representing 37 annual anomalies of the values of the sea ice extension for the 37 values on 21 July); occurred in 2004. DS only has CPs in the period from July to October. These CPs occurred mostly in 2004, the first part of September in 2006, and the last week of October in 2003. The results of both the BinSeg and PELT methods (results not shown) coincide for every day except for 17 September (2004 based on AMOC and 2006 for the other methods). The red bands in figure 7 portray the

CPs in the MS data set (July–October). For instance, for July monthly anomaly series (size of each series: 37 data points, representing 37 annual anomalies of the values of the sea ice extension for the 37 values in the average monthly July); occurred in 2004. These CPs occurred in 2004 for July and September, a little earlier (2001) for August, and a little later (2005) for October. Both the BinSeg and PELT methods coincide in every month except for October (2005 based on AMOC and BinSeg, and 2006 for PELT). The single green square corresponds to the CP in the 13,505-value series consisting of all days from 1

January 1980 to 31 December 2016 (ADS). The CP occurred on 22 September 2003, and was identified by the BinSeg, although not by PELT (the closest ones are 4 August 2002 and 26 January 2005). The single purple line represents the CP in the 444-value series consisting of all months from January 1980 to December 2016 (AMS). This CP occurred in October 2003. The results of both the BinSeg and PELT methods coincide for this CP, and it is the only one for both. Overall, the results in figure 7 suggest that 2003 is the most appropriate year for analysis of differences in moisture transport after and before a single

CP date. The average values for ADS before/after the CP were 0.27/−0.91 and for AMS were 0.41/−0.74. The analysis of changes in MTP for the multiple sub-periods identified by BingSeg and PELT would merit analysis, but it is out of the objective of this paper.

The Figure 8 portrays the differences between mean values of MTP until 2003 and mean values after 2003 for every source

region (figure 1c) for the AR, which includes continental and oceanic areas (figure 1a). The quantities in the plot result from averaging daily values of MTP. The statistical significance of the differences has been estimated by comparing daily values of MTP before and after the CP, and the sample size is large enough (30 x 23 years vs 30 x 13 years) to permit application of the Student *t*-test. The pattern of changes in MTP before and after the change point for the AR shows no significant changes in late winter and spring, a significant decrease in MTP in summer, and increased MTP in fall and early winter, with the exception

of October. The summer decrease is statistically significant for the contributions of Pacific and Siberian sources throughout the entire summer, for the Atlantic source in early summer, and for the North American source in late summer. The fall–early winter increase is statistically significant (red crosses) for the contributions of the North American source in three months (September, November, and December), for the Siberian and the Atlantic sources in two months (September and November;

and October and December, respectively), and for the Pacific source only in one month (November). As, according to figure 7 mainly for DS and MS, 2004 could also be interpreted as the main change point, we tested results of changes in MTP by changing 2003 by 2004 with almost identical results (not shown).

These results are coherent with any of the mechanism referred in the introduction. So, i) a lower MTP in early summer (as occurred since 2003) is consistent with lower precipitation as snowfall which would result in a decreasing in the surface albedo and thus increasing melt *(Cheng et al., 2008);* ii) a lower MTP in late summer (as occurred since 2003) is consistent with less probability of occurrence of rainfall storms with possible flooding over the ice which would favor the formation of superimposed ice and consequently is consistent with increasing melt; iii) a higher MTP in early fall (September) (as occurred since 2003) is consistent with higher precipitation as rainfall, something generally related to sea ice melt; and iv) a higher MTP in late fall and early winter (as occurred since 2003) is consistent with higher precipitation as snowfall, enhancing thermal insulation and thus reducing sea ice growth in *(Leppäranta, 1993).* The rigorous checking of these implications merits further analysis but it is out of the scope of this manuscript, since it would imply to know details over the precipitation form (snow or rain) for the different Arctic regions with good temporal and geographical resolution, and even to analyze specific precipitation episodes to know if these are responsible for flooding or not.

However, because Arctic ice cover is extensive in geographical domain covering the subregion affected by very different atmospheric circulation patterns, this pattern of change in MTP could be non-homogeneous for the entire AO and its subregions. Finer-scale analysis can be done by restricting the analysis to oceanic areas only (the 13 Arctic oceanic subregions and the whole Arctic Ocean defined as the sum of these regions, figure 1b). The differences between mean values of MTP until 2003 and mean values after 2003 for every source region for the AO (figure 9 top left) is quite similar to the AR without significant changes in late winter and spring, with decreased MTP in summer and increased MTP in fall and early winter, now also including October. Small differences are observed in the contributions of each source to the change of MTP toward the AO, with the Pacific source becoming much less important for the summer decrease and more important in the fall increased. The consistent increment is especially remarkable after the change point of MTP for all the moisture sources in September, the month when extension of Arctic sea ice is lowest.

The analysis by subregion allowed us to identify which subregions contributed most to change in MTP in the AO. As in the case of the AR, the statistical significance of the differences (Table 1) has been estimated by comparing daily values of moisture transport before and after the CP. The diminished contribution of the Atlantic source to the AO in June and July is mainly attributed to the reduction of transport to Greenland; this decrease is not observed in August because of the compensation of the decrease to Greenland by the increase to Baffin. A moderate increase in October and a small decrease in December were also attributed to changes occurring in Greenland. The slightly decreased contribution of the Pacific to the AO in summer was caused by the compensation of the decreased contribution of Bering (strong decrease) with a strong increase in the contribution of Okhotsk, with some influence of other regions such as Beaufort (decrease) and East Siberian (increase). There is greater concordance in fall and winter, with generally slight increments, which resulted in a small increase in the contribution of the

Pacific to the AO. The greatly decreased contribution of the Siberian source to the AO in summer is attributed mainly to the strong decrease of MTP to the Central Arctic and Laptev, and the generally slight increment of the contribution of the Siberian source for these regions in fall and early winter (with the exception of October); these changes result in similar behaviour for the AO. The different changes of the contribution of the North American source to the AO in July (increase) and August (decrease) reproduce the common changes observed in the contributions to Baffin and Hudson. The slight increase in contribution to the AO in September and October again is attributed to compensation of the increased to Hudson with a strong decrease in September to the Canadian Islands and light decreased in October to Baffin and Barents; the increased contribution of this source to the AO in December is the result of increased contribution to the Central Arctic and Barents, partially compensated by slight decrease to Hudson and Baffin.

### 3.3 Checking the Lagrangian results by analysing changes in vertical integrated moisture fluxes and atmospheric circulation patterns after and before Arctic sea ice change points

It is useful to check the results derived from the Lagrangian approach to estimate MTP with other Eulerian approach such as the computation of vertical integrated moisture flux (VIMF). Supplemental figure S1 shows the climatological VIMF by month in the period from June to December (left panel) and the difference between the periods after the CP and before the CP for the zonal component (central panel) and the meridional component of VIMF (right panel). This methodology cannot estimate specific changes in the MTP from each source to each sink as the more sophisticated Lagrangian approach does, but it is able to show whether patterns are compatible with the identified changes. For instance, in June (figure 10), the Atlantic source provides moisture for precipitation in the subregion of the Arctic Ocean named Greenland in our study, as revealed by the flux vectors in the top panel. However, the track of moisture from the source to the sink is clearly hindered for the period after the CP relative to the period before the CP, as revealed by negative values of the zonal component (blue colours in middle figure 10) in the band 45–60ºN latitude (less moisture transported from west to east, the direction of the source–receptor path) and negative values of meridional winds (blue colours in bottom figure 10) in the band 20–60ºW longitude (less moisture transported from south to north, the direction of the source–receptor path). We have done this analysis for each month, and for source and sink areas, and the patterns can explain all the significant results found in the Lagrangian analysis with almost absolute agreement (figure S1).

An additional demonstration of the robustness of these results can be derived by analysing the changes in atmospheric circulation responsible for changes in moisture transport. Changes in MTP may be related either to alterations in moisture sources caused by changes in circulation patterns, or to changes in the intensity of the moisture sources because of enhanced evaporation, or to a combination of these two mechanisms. At a daily scale, changes in intensity are negligible, but changes in circulation may be significant. Thus, to analyse whether there are important differences in MTP from sources to sinks associated with different circulation patterns, and how such differences could affect MTP in the AR, we grouped individual days into four circulation types using the methodology developed by *Fetttweis et al. (2011)* and explained in the Data and

Methods sections. We have separated the analysis into four sectors centred on the four sources of moisture to the AR to filter out annular effects, as we are interested in regional influences.

Supplemental figure S2 shows the summer and fall types of circulation centred on the four source areas (Atlantic, Pacific, Siberia, and North America), with each individual map representing the anomalies of geopotential height at 850 hPa (Z850)
for the four different classes found (from CTC1 to CTC4, with the percentage of days grouped). The average MTPs for each class before and after the CP are summarised in Table S2, whereas changes in the frequency of each class before and after the CP are shown in Table S3 (statistical significant changes were calculated using a z-test to compare two sample proportions - *Sprinthall, 2011*-). One of these sectors and one season can serve us as an example of these results: the Atlantic during summer (figure 11). The anomalies of geopotential height at 850 hPa (Z850) for the four different classes found (figure 11 top) show
patterns that resemble the known teleconnection patterns in the region (*Barnston and Livezey, 1987* and http://www.cpc.ncep.noaa.gov/data/teledoc/telecontents.shtml). The term teleconnection pattern alludes to a recurring and persistent large-scale pattern of geopotential height and circulation anomalies over large geographical areas and with strong influence on meteorological varaibles including rainfall. Thus, CTC1 has a zonal dipolar structure that resembles the positive phase of the East Atlantic pattern (a north-south dipole of anomaly centers-positive in the southern one-spanning the North
Atlantic), closely related to enhanced precipitation on Greenland; CTC2 is slightly similar to the negative phase of the East Atlantic/western Russia, (negative height anomalies located over Europe and northern China, and positive height anomalies located over the central North Atlantic and north of the Caspian Sea), associated with enhanced precipitation on Barents; CTC3 resembles the negative phase of the North Atlantic Oscillation (an above-normal heights across the high latitudes of the North Atlantic and below-normal heights over the central North Atlantic and western Europe), related to diminished precipitation on
Greenland and enhanced precipitation on Barents, and CTC4 is similar to the positive phase of the Scandinavian pattern (positive height anomalies over Scandinavia and western Russia) associated with diminished precipitation on Barents. The middle panel of figure 11 shows the average MTP for each class in the summer months from the Atlantic source to the AO (dark blue line) and the two dominant subregions, Barents and Greenland (light blue and orange lines, respectively). CTC1 is the circulation type responsible for the highest MTP in the three summer months, and CTC3 and CTC4 are responsible for the
lowest MTP, depending on the month. The change of frequency of these classes representing circulation types before and after the CP (Figure 11 bottom) shows a decrease of the days belonging to CTC1, with the highest MTP from 40 to 25.4 days, and an increase of days belonging to CTC3 (from 9.58 to 15.15) and CTC4 (from 4.62 to 5.85), the two classes with the lowest MTP. These results are absolutely consistent with our Lagrangian results of changes in MTP. A similar analysis to the one for this sector, moisture source region, and season can be done for all the significant results found in the Lagrangian analysis with
very good agreement.

## 4 Concluding remarks

This study shows that a drastic Arctic sea ice decline occurred in 2003 and that this decline was accompanied by a change in the moisture transport from the main Arctic moisture sources, which then results in precipitation over the Arctic (MTP). The pattern of change consists of a general decrease after vs before 2003 in the moisture transport in summer and an increase in

fall and early winter, with different contributions depending on the moisture source and ocean subregion. This pattern *of change in the moisture transport* is not only statistically significant but also consistent with Eulerian flux diagnosis, changes in circulation type frequency, and *any of the* known mechanisms affecting snowfall or rainfall on ice in the Arctic. The consistent increment after the CP of MTP for all moisture sources in September, the month when extension of Arctic sea ice is lowest, is particularly remarkable. These results suggest that ice-melting at a multiyear scale is favoured by a decrease in

moisture transport in summer and an increase in fall and early winter.

The results of this paper also reveal another important conclusion: the assumed and partially documented enhanced poleward moisture transport from lower latitudes as a consequence of increased moisture from climate change *(Zhang et al., 2013)* has not been simple or constant in its links with enhanced Arctic precipitation throughout the year in the present climate. Major

moisture sources for the Arctic did not provide more moisture for precipitation to the Arctic in summer after 2003, the change point (CP) for Arctic sea ice, than before, although they did provide more moisture in fall and early winter. Because the enhanced Arctic precipitation projected by most models for the end of the century *(Bintaja and Selten, 2014)* is partly attributed to enhanced poleward moisture transport from latitudes lower than 70ºN *(Bintanja and Andry, 2017),* where the major sources studied herein are located, our results raise questions of whether this change has occurred so simply in the current climate, and

these questions merit further study.

### Acknowledgements

The authors acknowledge funding by the Spanish government within the EVOCAR (CGL2015-65141-R) project, which is also funded by FEDER (European Regional Development Fund). The data used are listed in the references, tables and

25 supplements.

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

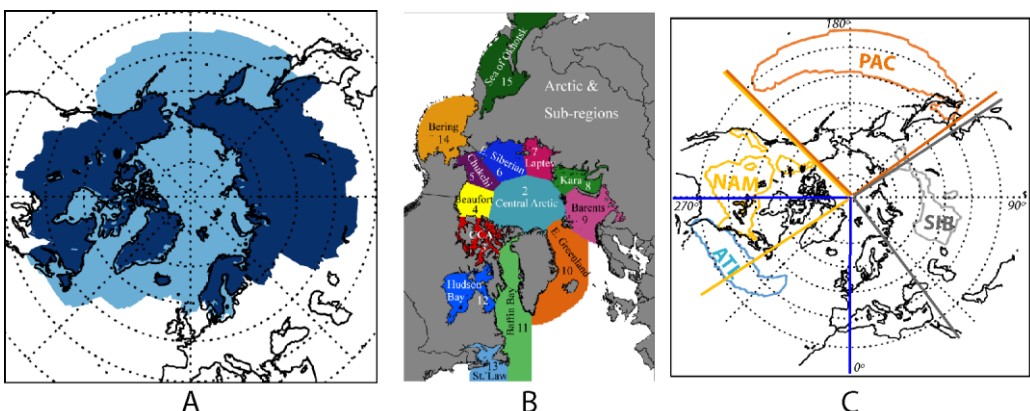

Figure 1. A) The Arctic region (AR) included in the present work defined following the definition of Roberts et al. (2010). B) The
Arctic Ocean (OA) and its 13 subregions as described by Boisvert et al. (2015). C) Major moisture sources for the Arctic as detected
by Vazquez et al. (2016). The coloured longitudinal lines mark the areas used for the types of circulation analysis in figure 11.

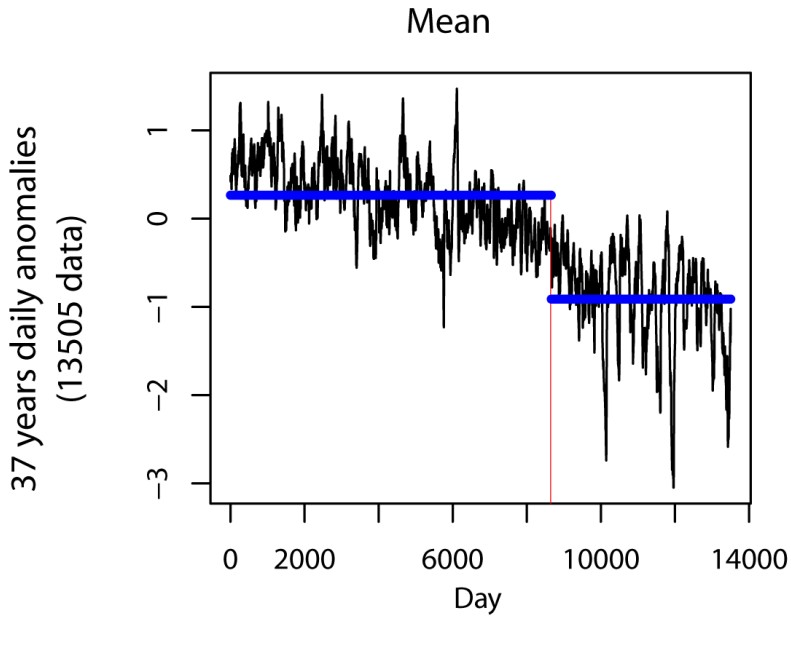

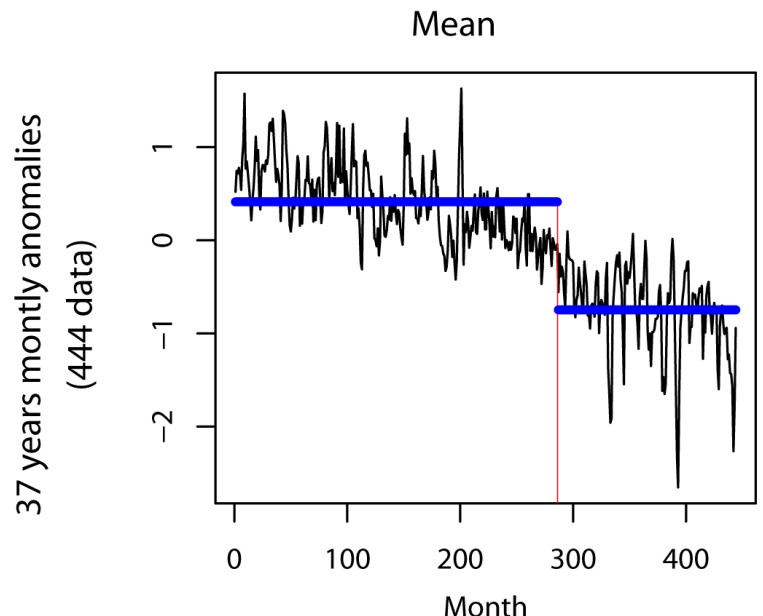

**Figure 2. Example of change point detection in mean for two series and one method, AMOC. Top plot represents the change points in ADS series. The two horizontal lines represent the mean of the values before and after the change point identified by the AMOC method (8660th day - 22th September 2003). Those means are 0.27 and -0.91, respectively. Bottom plot represents the change points in AMS series. The two horizontal lines represent the mean of the values before and after the change point identified by the AMOC method (286th month - October 2003). Those means are 0.41 and -0.74, respectively.**

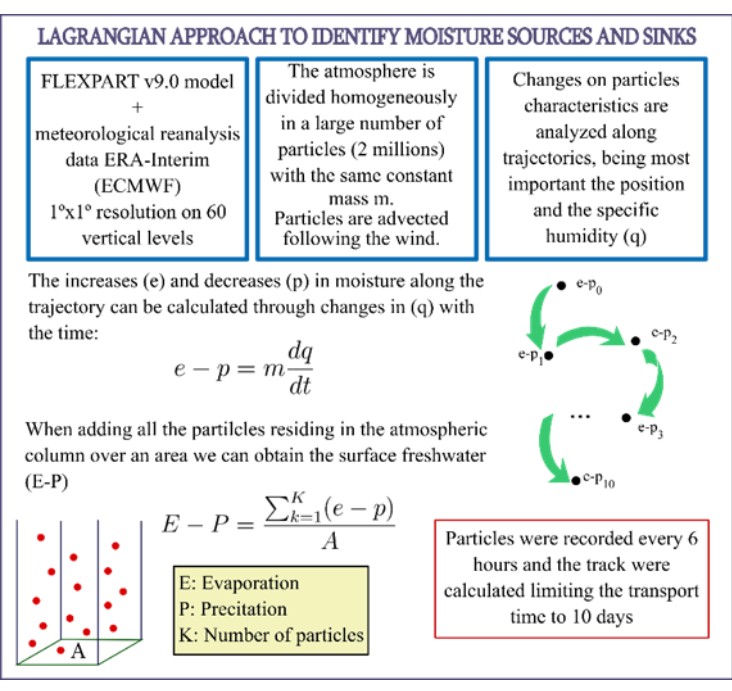

**Figure 3. A scheme of the Lagrangian approach used to estimate moisture transport.**

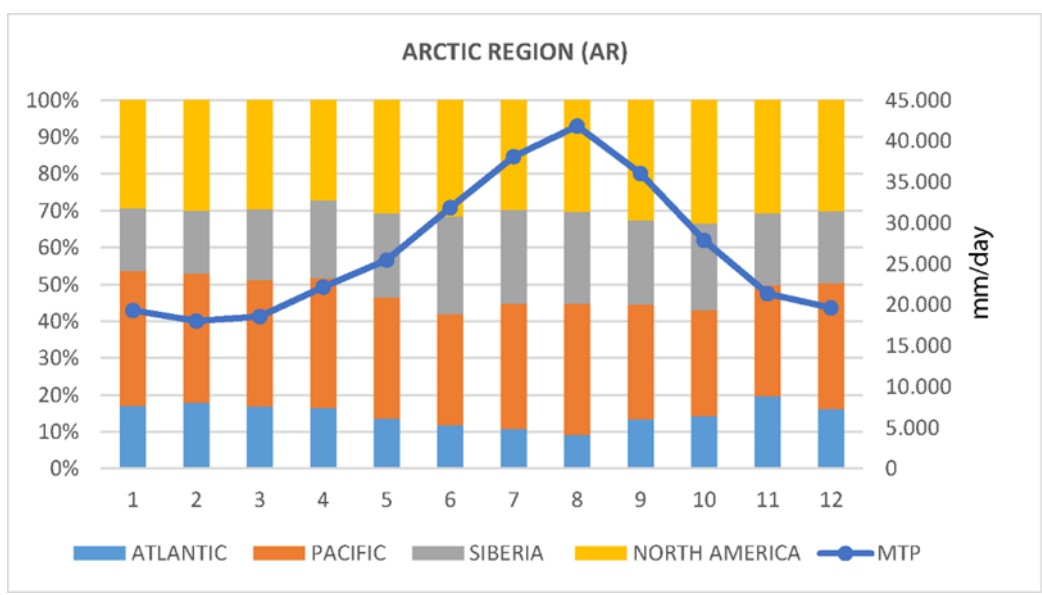

**Figure 4. Seasonal cycle of moisture transport for precipitation (MTP) to the Arctic region (AR) from the four major sources (Atlantic, Pacific, Siberia, and North America). Values at the right represent absolute values of transport, and those at the left indicate the relative contribution of each source by percentage.**

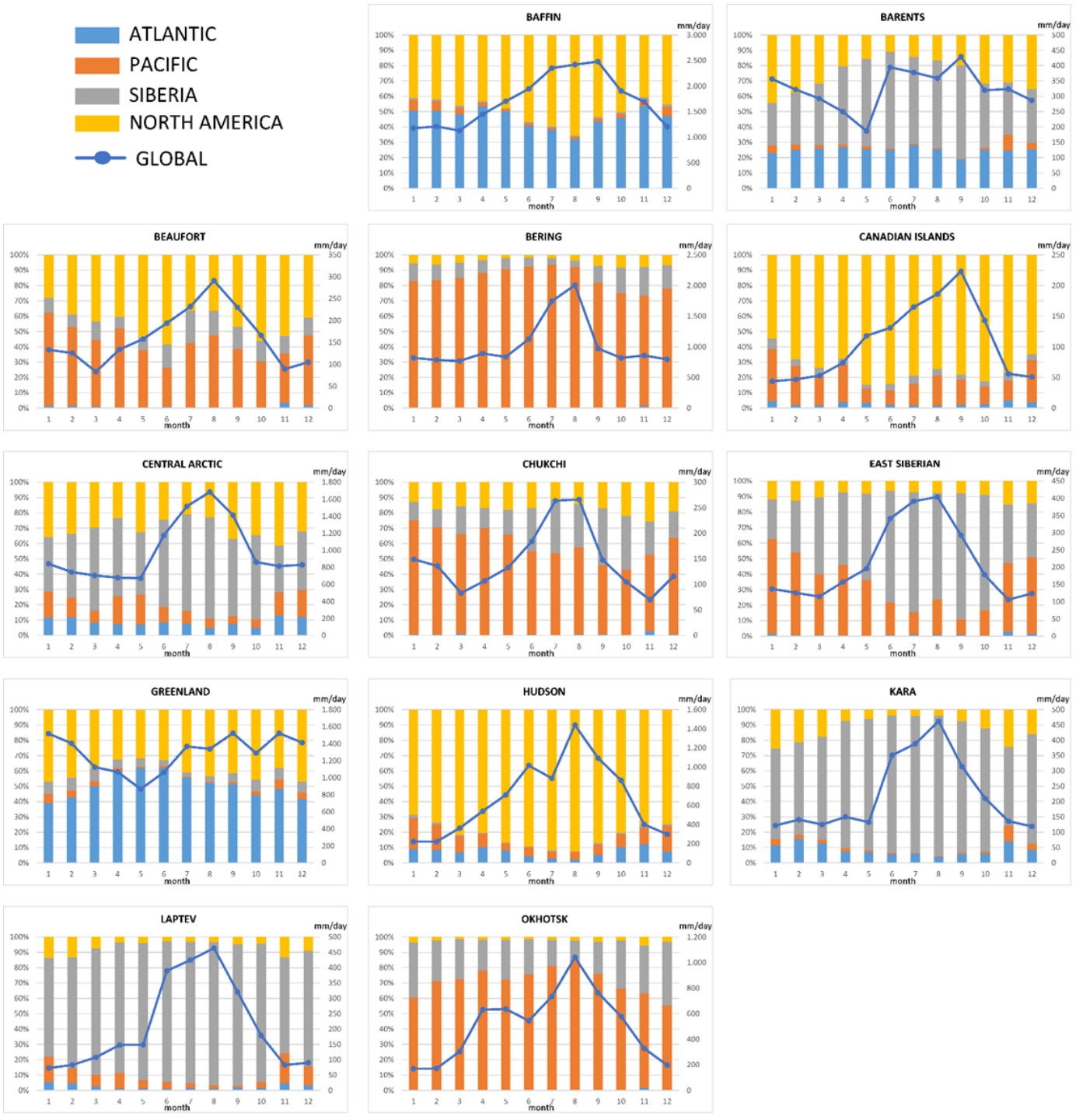

**Figure 5. Same as in figure 3, but for each of the 13 subregions of the Arctic Ocean (AO).**

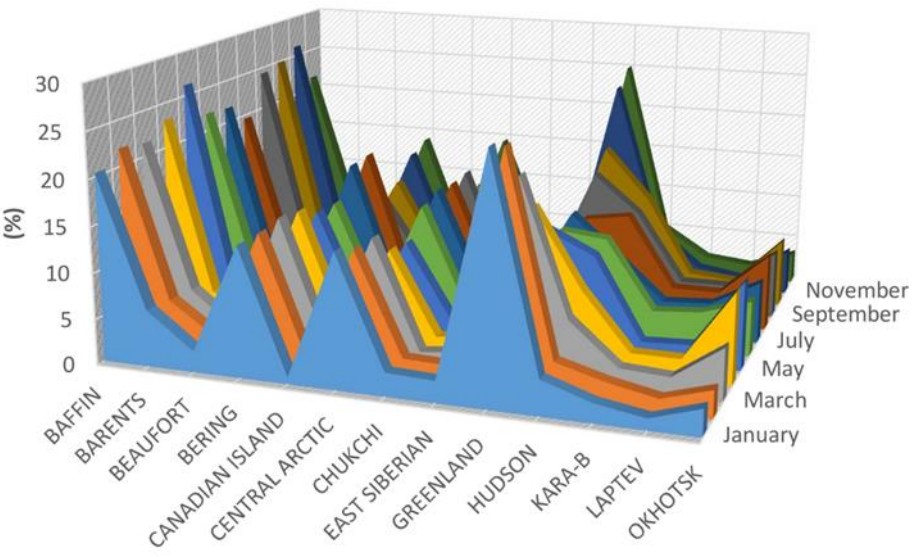

5 **Figure 6. Percentage of moisture transport for precipitation to the whole Arctic Ocean by subregion and month.**

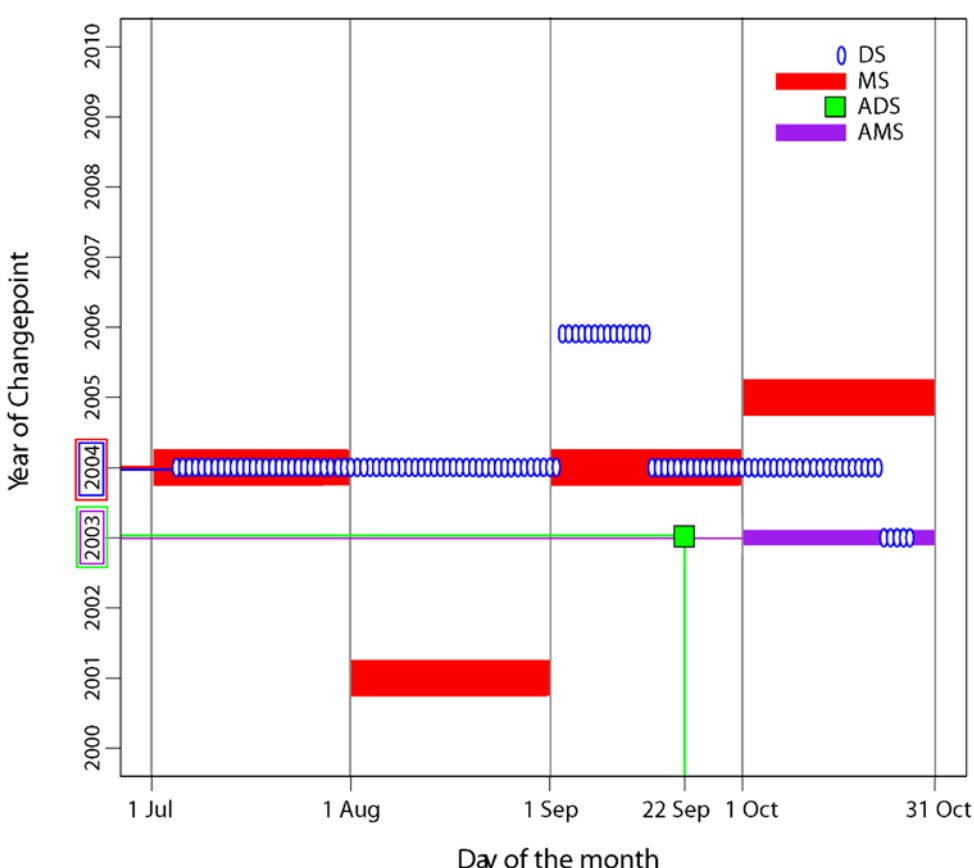

5   **Figure 7. Summary of the identified change points (CPs) in means identified using the AMOC method with the four series of ice extent anomalies for the whole Arctic (DS, MS, ADS, and AMS) from 1980 to 2016. The blue points refer to the change points in DS; the red lines portray change points in the MS (July–October). The green square corresponds to the change point in the ADS, and the purple line represents the change point in the AMS.**

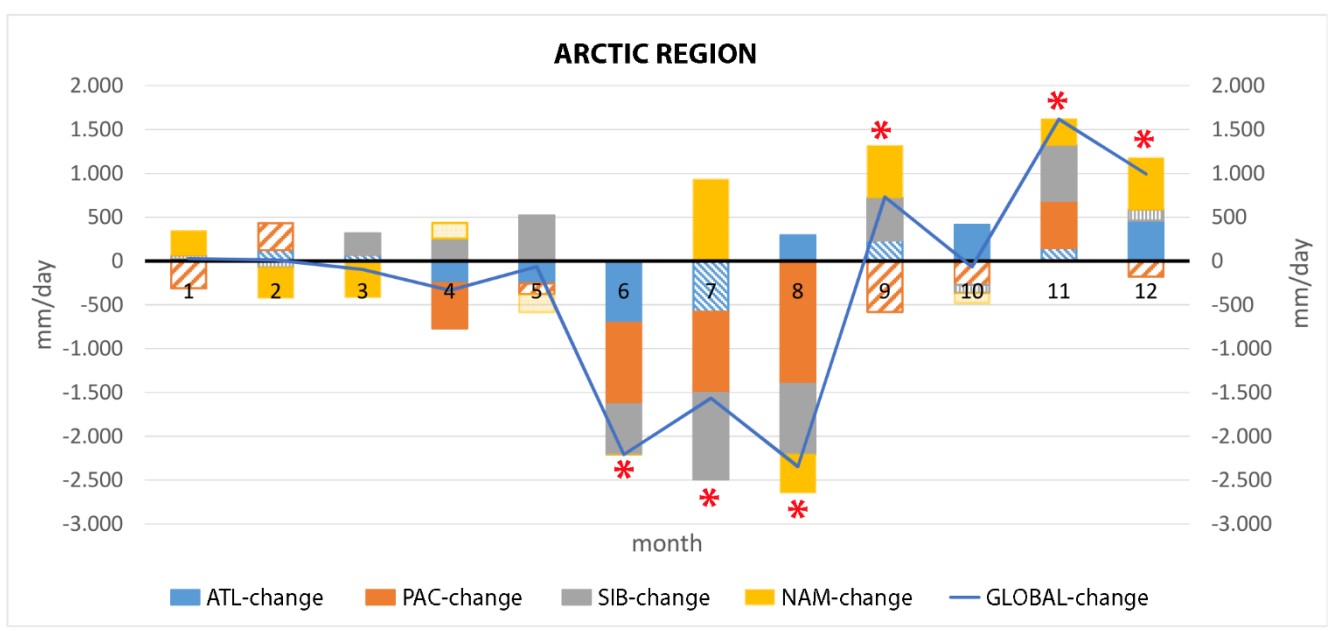

**Figure 8. Differences between mean values of Moisture transport for precipitation (MTP) until 2003 and mean values after 2003 for every source region. Filled bars show those differences that are statistically significant at the 95% confidence level for decreases after the CP. Statistical significance of the differences in total MTP (sum of the four sources) is displayed with a red asterisk.**

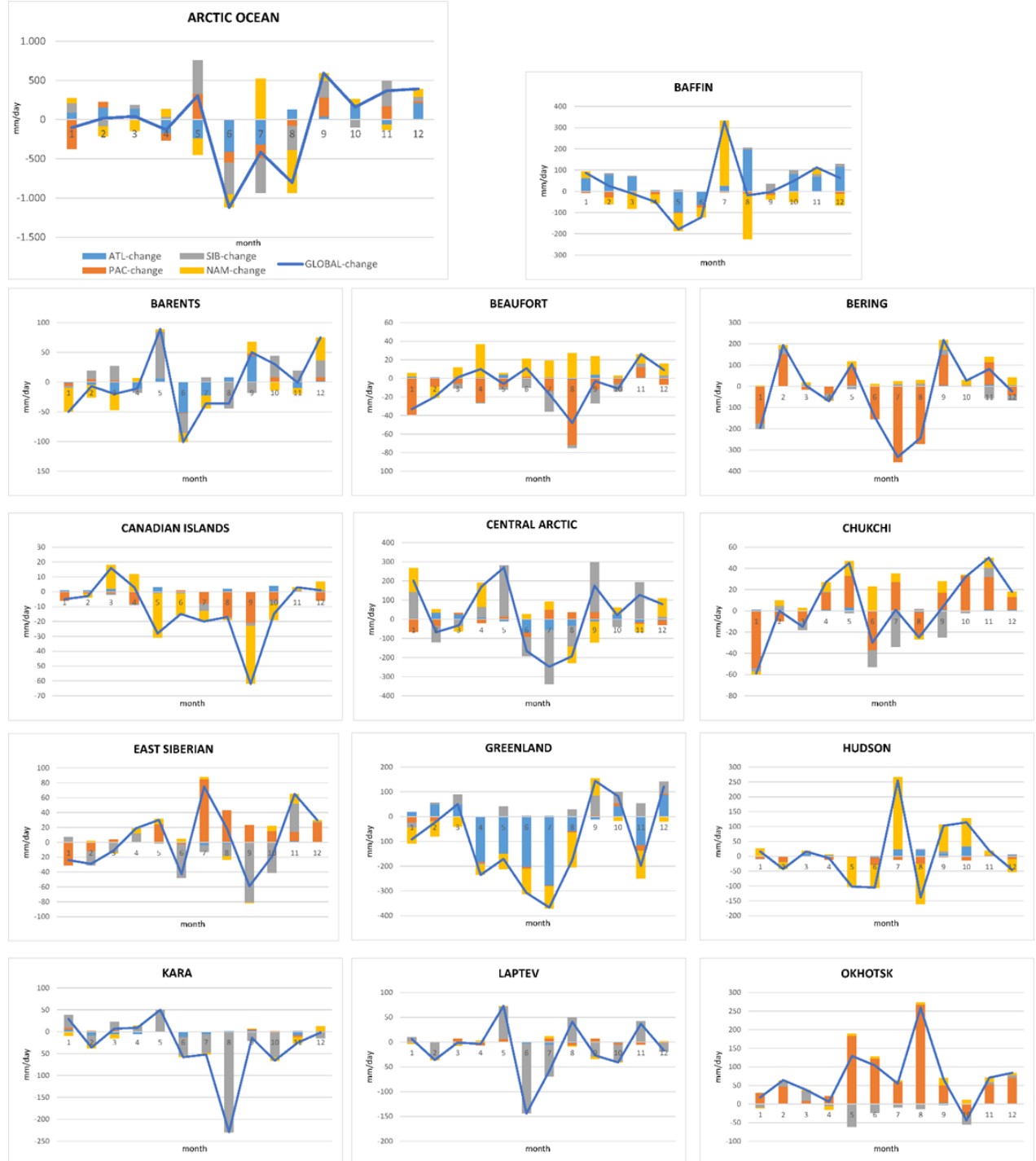

**Figure 9. Differences between mean values of Moisture transport for precipitation (MTP) until 2003 and mean values after 2003 for every source region for the Arctic Ocean (AO) and its 13 subregions. Statistical significance of the differences is displayed in Table I.**

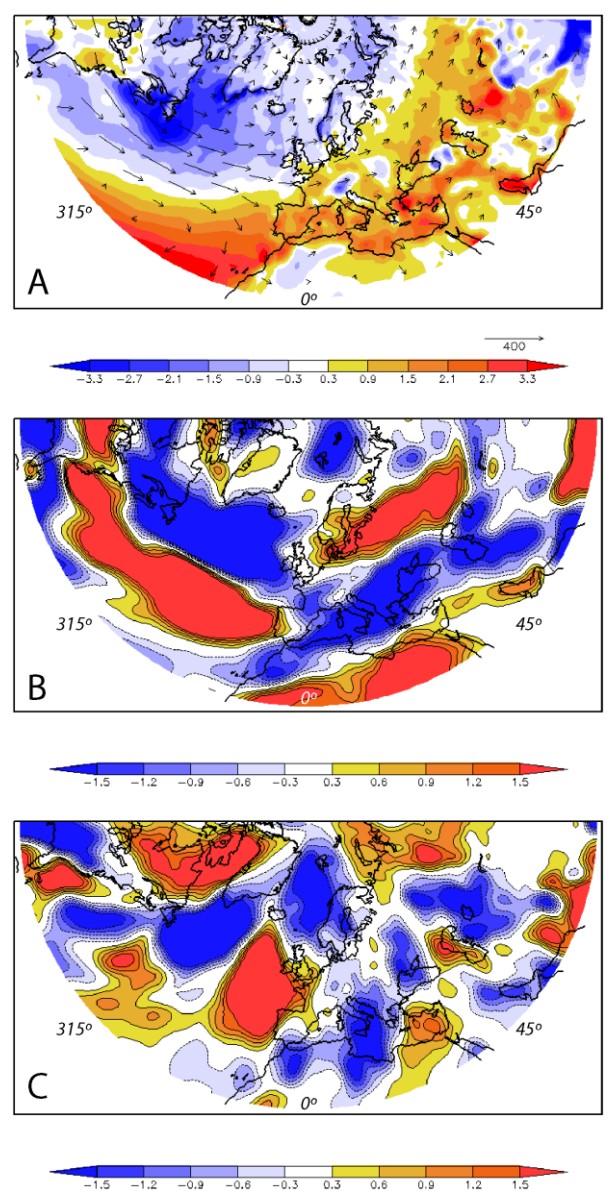

**Figure 10. A)** The climatological vertical integrated moisture flux (VIMF) (vector, kg/m/s) and its divergence (shaded, mm/yr) in June for the European sector, **B)** the difference between the periods after vs before the CP for the zonal component of VIMF, and **C)** as for b but for the meridional component of VIMF.

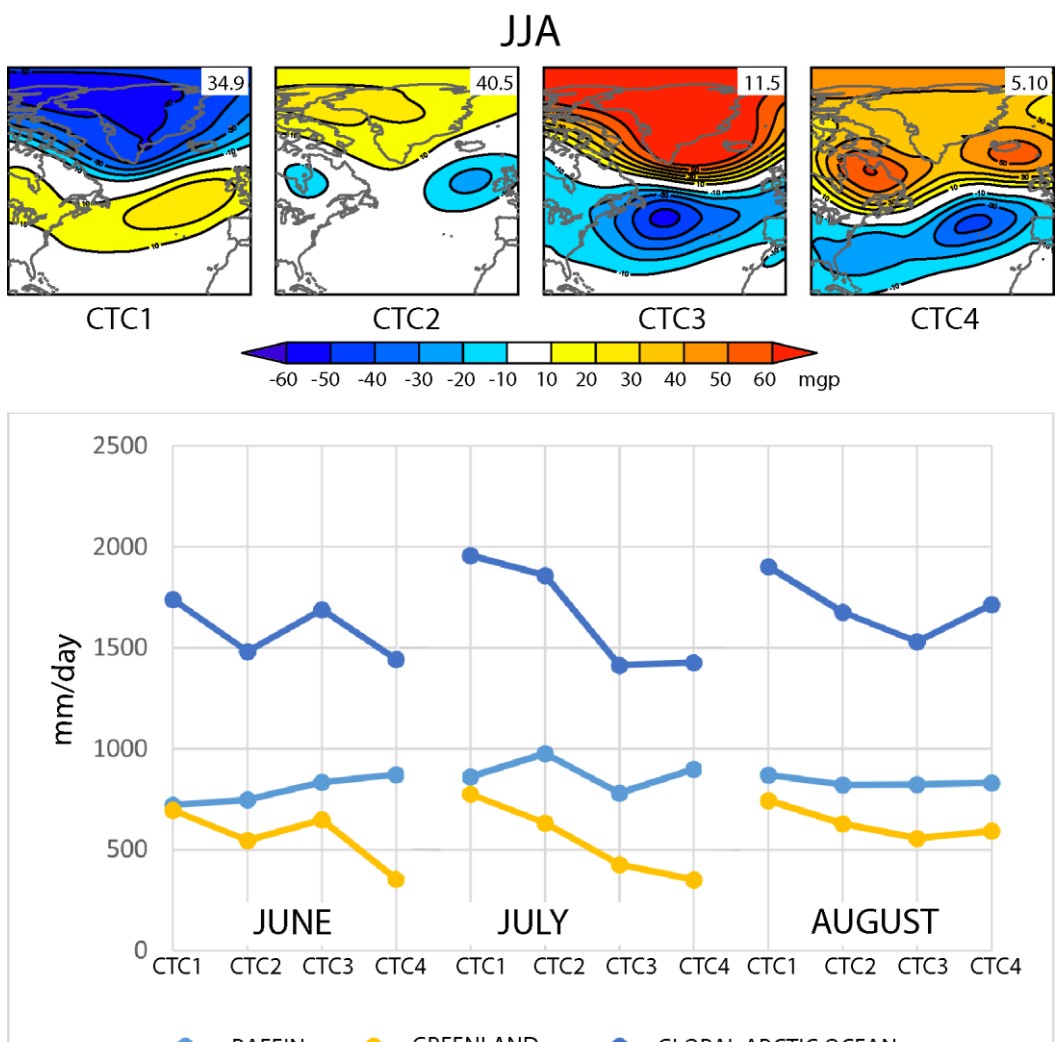

| | CTC1 | CTC2 | CTC3 | CTC4 |
|---|---|---|---|---|
| before | 40 | 37,79 | 9,58 | 4,62 |
| after | 25,54 | 45,46 | 15,15 | 5,85 |

**Figure 11. (top) Anomalies of geopotential height at 850 hPa (Z850) for the four types of circulation centred in the Atlantic sector in summer (classes CTC1 to CTC4), and the percentages of days grouped in each class. (middle) Average moisture transport for precipitation (MTP) for each class and summer months from the Atlantic source to the Arctic Ocean (AO), and the two dominant subregions, Barents and Greenland. (bottom) Change of frequency of these circulation types (classes) before and after the change point.**

| ATLANTIC SOURCE | Jan | Feb | Mar | Apr | May | Jun | Jul | Aug | Sep | Oct | Nov | Dec |
|---|---|---|---|---|---|---|---|---|---|---|---|---|
| Baffin | | | X | | X | | | X | | | | X |
| Barents | | | X | | | X | X | | X | | | |
| Beaufort | X | | X | | X | | | | X | X | X | X |
| Bering | | X | X | | X | | X | | X | | X | X |
| Canadian | X | X | X | | X | | | X | | X | | X |
| Central Arctic | | X | X | | | X | X | X | | X | | |
| Chukchi | X | | | | X | | | X | X | | X | X |
| East Siberian | | X | | | X | X | X | X | | | X | |
| Greenland | | | | X | X | X | X | | | | X | X |
| Hudson | | | X | | | | X | X | | X | | |
| Kara | X | X | X | X | | X | | | | | X | |
| Laptev | X | X | | | | X | X | X | | X | | X |
| Okhotsk | | X | | | X | | | X | X | | X | X |

| PACIFIC SOURCE | Jan | Feb | Mar | Apr | May | Jun | Jul | Aug | Sep | Oct | Nov | Dec |
|---|---|---|---|---|---|---|---|---|---|---|---|---|
| Baffin | | X | | X | | X | | X | | | | |
| Barents | X | | | | | | | | X | X | | X |
| Beaufort | X | | | X | | | | X | | | X | |
| Bering | X | X | | | | X | X | X | X | | X | |
| Canadian | X | | | X | | | X | X | X | X | | X |
| Central Arctic | X | X | | | | | | | | | | |
| Chukchi | X | | | | X | X | | | | X | X | |
| East Siberian | X | | | | X | | X | X | X | X | | X |
| Greenland | X | X | | X | | X | | X | | X | | X |
| Hudson | | X | | X | | X | X | X | | X | | X |
| Kara | X | X | X | | | | | | X | X | X | |
| Laptev | | | X | X | X | | | X | X | X | X | |
| Okhotsk | X | X | | | X | | X | X | | | X | X |

| SIBERIAN SOURCE | Jan | Feb | Mar | Apr | May | Jun | Jul | Aug | Sep | Oct | Nov | Dec |
|---|---|---|---|---|---|---|---|---|---|---|---|---|
| Baffin | | X | X | | X | | | X | X | X | X | X |
| Barents | | | X | | X | | | X | | X | | X |
| Beaufort | | | X | | X | X | X | | X | X | X | |
| Bering | X | X | | X | X | | | | X | | X | X |
| Canadian | | | X | X | | | X | | X | X | X | |
| Central Arctic | X | X | | | | | X | | X | | X | |
| Chukchi | | | X | | | X | X | | X | | X | |
| East Siberian | | X | X | | | X | | | X | X | X | |
| Greenland | X | | X | | X | | | X | X | X | X | X |
| Hudson | X | X | X | | | | | X | X | X | X | X |
| Kara | X | X | X | | X | | | X | | X | | X |
| Laptev | X | X | | | X | X | X | | | X | X | X |
| Okhotsk | X | X | X | | X | X | | | | X | | |

| NORTH AMERICA SOURCE | Jan | Feb | Mar | Apr | May | Jun | Jul | Aug | Sep | Oct | Nov | Dec |
|---|---|---|---|---|---|---|---|---|---|---|---|---|
| Baffin | | | X | | X | | X | X | | | | X |
| Barents | X | | X | | | X | X | | X | | | X |
| Beaufort | | X | X | X | | | X | X | | | X | X |
| Bering | | X | | | X | X | X | | X | X | X | X |
| Canadian | | | X | X | X | X | | | X | | | |
| Central Arctic | X | | | X | | | | X | X | | | X |
| Chukchi | | | | X | X | X | X | | X | | X | |
| East Siberian | | | | X | X | | | | | X | X | |
| Greenland | | | | | X | X | X | X | | | X | |
| Hudson | | X | | | X | | X | X | X | X | | X |
| Kara | X | X | X | | | X | | | | X | X | X |
| Laptev | X | | X | | | | X | | X | | | X |
| Okhotsk | | X | X | | X | X | | | X | X | X | X |

**Table I.** Summary of the statistical significance of the MTP differences estimated by comparing daily values of moisture transport before and after the CP, based on the Student t-test. Red crosses indicate statistical significance at the 95% confidence level for increases after the CP; blue crosses indicate statistical significance at the 95% confidence level for decreases after the CP.