# Peer review of "The pattern of long-term changes in the moisture transport for precipitation with Arctic sea ice melting"

_Earth System Dynamics, 2017_

## Referee Comment (RC1) · Anonymous Referee #1 · 24 Jan 2018

"The perfect pattern of moisture transport for precipitation for Arctic sea ice melting" (by Gimeno-Sotelo et al.) studies the changes in the patterns of moisture transport before and after 2003, identified as the main change point in Arctic sea ice extension series. Moisture transport decreases in summer and is enhanced in autumn and early winter. These results are shown to be consistent with other approximations to the problem and provide a reasonable explanation about the observed reduction in sea ice extension.

I find this paper to be an interesting addition to previous knowledge about mechanisms related to Arctic sea ice reduction in the last decades. But, before accepting this paper for publication, some new explanations and discussions have to be added and some figures and tables should be corrected or improved.

Before any comments about the text or the figures, I have to say that I do not feel very

comfortable with the title of the paper. In the paper, it is shown that there has been a modification in seasonal transport of moisture toward the Arctic and in the prevailing circulation patterns in the Arctic and that those new conditions lead to a reduction in sea ice extension. But I do not understand why the new pattern is considered as 'perfect'. Is the new pattern the one that maximizes the reduction in sea ice extension (if that is what is called 'perfect')? I would suggest to change the title.

Some methodological and conceptual issues:

Section 2.2.3: '...To compute moisture transport for precipitation (MTP) from each source to each sink for the AO, the trajectories of particles from the moisture sources for the Arctic (AR) were followed forward in time from every source region detected by Vazquez et al. (2016) (figure 1c).' I find that further discussion is needed about this sentence (and paragraph) and what it implies. It is difficult for me to understand why those (and only those) particles are tracked. In fact, my interpretation is that source regions are not source regions anymore since authors follow 10 days into the future all the particles within those regions, had them gained water vapor within those source regions or not. Thus, water does not necessarily come from those regions and they stop being 'source regions'. In addition, not all the precipitated water comes from those 'sources', so, what happens with other particles that produce precipitation but were not within those 'source regions' ten days before precipitation? Another question, are there enough 'particles' to properly characterize what happens with the smallest sub-regions defined in figure 1b (I am thinking in the results presented in figure 8)? Section 2.2.3 (should be 2.2.4): 'circulation types' are identified for four sections selected 'according to the positions of the major sources of moisture'. What are the sizes of those sections? Is there a minimum recommendable size? Is the method used to identify 'circulation types' sensible to the area selected? Are your results robust if you modify (nut much) those sections? Some of those sections share some common areas, does it affect to the latter interpretation of the circulation types? In addition, it would be advisable to identify those four sections in figure 1c.

Results:

Section 3.2: Authors state that results in figure 6 suggest that 2003 is the most appropriate CP year. I do not find it so obvious. DS and MS series (figure 6) suggest that 2004 would be a better selection. Have authors tested if selecting one year or the other produces any difference? And, when writing about CPs, some explanation about change points identified using BinSeg and PELT should be provided. It is not clear in the text if more than one CP has been identified using those methods nor the implications that the existence of more than one CP in the SIE series would have in the interpretation of the results of this paper.

Some additional comments and typos:

P7, paragraph describing figure 7: It is not explained anywhere that figure 7 includes the differences between mean values of MTP until 2003 and mean values after 2003 for every source region (this is my interpretation of what is represented in figure 7). The caption of figure 7 doesn't include this information either. Same comments can be applied to figure 8.

I would suggest to re-plot figures 7 and 8 in order to include the information from the table in figure 7 and from table 1. It would be as easy as to plot with a thick (or filled) bar those differences that are statistically significant and with a thinner (empty) bar those differences that are not. In addition, plotting with a thicker line the horizontal bar indicating the 0 mm/day level would help to notice which sources increase/decrease their MTP contribution. Finally, no information about the statistical significance of the differences in total MTP is provided anywhere (again, it could be indicated by using a continuous or discontinuous line)

It would be easier to follow the comments in the text if Figure S1 and figure 9 included some labeled meridians (or at least, some longitudes in the outer area of the maps).

P2l3: 'ong these effects. . .' I guess it should be 'Among these effects. . .'

P3l14 (p3l16): 'vs' -> '-'

P3l26: delete 'uses' or 'employs'

P5l12: '. . .using an applied methodology. . .' -> '. . .using a methodology. . .'

P6l7: delete '(Greenland)' in '. . .to fall and winter (Greenland)'

P6l8: '. . .with the a dominating closest source' -> '. . .with the closest source dominating'

P6l15: 'represent' -> 'represents'

P6l21: lesser

P7l20: '. . .MTP could not be homogeneous. . .' -> '. . .MTP could be non-homogeneous. . .'

P9l29: Again, I do not know why this is a 'perfect pattern of MTP for Arctic sea ice melting'. It is shown that it favors the melting, but. . .

P10l2: 'it is clear beyond doubt. . .' Is it?

Table S2: Are changes in the frequency of each class statistically significant?

---

## Referee Comment (RC2) · Anonymous Referee #2 · 6 Mar 2018

Review of the manuscript "The perfect pattern of moisture transport for precipitation for Arctic sea ice melting" by Gimeno-Sotelo, Nieto, Vazquez and Gimeno submitted to ESD.

The manuscript addresses an important question about the potential role of poleward moisture transport and associated changes precipitation in the recently accelerating Arctic sea ice decline. In order to identify the tipping points in the Arctic sea ice extent, the authors apply a statistical approach of computing the so called "change points" and focus on the year 2003 as the most important change point further analyzing changes in the moisture transport and atmospheric circulation patterns before and after this point. The moisture transport patterns are identified first by a Lagrangian approach estimating E-P along 10-day forward trajectories initiated from four sub-Arctic regions

[Figure]

- and integrated over different Arctic regions. The authors use the moisture source region identified in a previous study as primary climatological moisture source regions. Particularly, the detected changes since the 2003 tipping point include a decrease in the (E-P) in summer and increase in fall and early winter (except for October).

I believe that combing these different methodologies can give new insights into the processes behind the observed sea ice extent changes. However, I find that conceptually the manuscript in its present stage misses to convey its message and needs major revisions before it can be considered for publication in ESD.

First, I find the title confusing and even misleading. Which perfect pattern is meant? Also, the statement "moisture transport for precipitation for Arctic sea ice melting" sounds rather awkward. I suggest to modify the title.

The authors present a detailed analysis with clear graphical representations, however methodology description is unfortunately too vague to appreciate and understand the results. I invite the authors to explain the concept of the change points in relationship to its application to the Arctic sea ice extent data. Further, it is not clear how the results shown in Fig 6 have been obtained and how these results can be interpreted. Section 3.2 text is very descriptive and lacks interpretation.

The methodology of the E-P analysis along the trajectories also needs to be better described. The moisture source regions are predefined from another study without any explanation - I invite the authors to explain the method in more details. The wording itself "moisture transport for precipitation" sounds confusing and has to be rephrased and better defined. The sentence explaining the methodology ("Then, we selected all particles losing moisture, $(e − p) < 0$, at the sinks (whole Arctic or any of the sub-regions), and by adding $e − p$ for all of these particles, we estimated moisture transport for precipitation from the source to the sink $(E − P) < 0$ at daily, monthly or yearly scales." ) needs more clarification with an extended explanation. This approach also doesn't imply that precipitation results exclusively from the moisture transport and local
moisture re-circulation can also contribute.

I find a lot of similarities of this manuscript with another article by a coauthor of the present article Vasquez et al 2017 (www.mdpi.com/2073-4433/8/2/32/pdf), which is not somehow in the reference list. Can the authors put this manuscript in context of Vasquez et al 2017 explaining the novelty of the results?

"We grouped individual days into four circulation types using the methodology developed by Fettweis et al. (2011 ) and explained in the Data and Methods sections." - in the Methods section the authors mention five (not four) circulation types and give no further explanations (which types, how they were defined...). Abbreviations used for the circulation types are not explained. What does it mean "the positive phase of the East Atlantic pattern" or " the negative phase of the East Atlantic/western Russia"?

I find that many statements in the Conclusions are not supported by the results. A major change seems to occur in 2003 - however unclear how this was obtained and what does it mean exactly (from the conclusions one can deduce that it means a drastic sea ice decline - I suppose the "change point" technique allows to detec that the mean SIE over the period after 2003 is significantly smaller than before). And the "perfect pattern of MTP for Arctic sea ice melting consists of a general decrease in moisture transport in summer and an increase in fall and early winter", as stated in the Conclusions section, refers to this year as I understand (no longer mentioned in the Conclusions). "This pattern is not only statistically significant but also consistent with Eulerian flux diagnosis, changes in circulation type frequency, and known mechanisms affecting snowfall or rainfall on ice in the Arctic." - which other known mechanisms affecting precipitation the authors refer to? "it is clear beyond doubt that an increment in moisture transport during this month favours ice melting, regardless of the source of moisture." - how is that clear? September's increase in MTP according to the methodology used here (if I understood correctly) means increased local precipitation vs evaporation (not necessarily increased moisture transport), and its impact to the SIE has not been established in this study. There are previous studies showing that the linkage can be the other way

around - that precipitation has increased because of the decreased sea ice extent (eg, Bintanja, R. & Selten, F. M. Future increases in Arctic precipitation linked to local evaporation and sea ice retreat. Nature 509, 479–482 (2014).).

"Snowfall is the dominant (almost unique) form of precipitation during most of the year, with the exception of late summer." - there is frequent rain during summer (and not only later summer), especially in the peripheral Arctic regions. Even in the central Arctic rain can occur in the very beginning of the melt period (like during SHEBA, eg Perovich et al 2002).

" when precipitation is produced in the form of snowfall on sea ice, it enhances thermal insulation, and reduces sea ice growth in winter (Leppäranta, 1993), but increases the surface albedo, and thus reduces melt in spring and summer (Cheng et al., 2008). These phenomena justify the opposite change in moisture transport for fall and winter versus spring." - how can these phenomena justify any changes in moisture transport?

The manuscript has to be checked for language and consistency - there are many vague, incomplete phrases.

---

## Author Comment (AC1) · 16 Mar 2018

**Anonymous Referee #1 "The perfect pattern of moisture transport for precipitation for Arctic sea ice melting"**

Before any comments about the text or the figures, I have to say that I do not feel very comfortable with the title of the paper. I would suggest to change the title.

*The reviewer is right. We´ll change the title to something less confusing such as "The pattern of long-term changes in the moisture transport for precipitation with Arctic sea ice melting"*

Some methodological and conceptual issues: Section 2.2.3: '. . .To compute moisture transport for precipitation (MTP) from each source to each sink for the AO, the trajectories of particles from the moisture sources for the Arctic (AR) were followed forward in time from every source region detected by Vazquez et al. (2016) (figure 1c).' I find that further discussion is needed about this sentence (and paragraph) and what it implies. It is difficult for me to understand why those (and only those) particles are tracked. In fact, my interpretation is that source regions are not source regions anymore since authors follow 10 days into the future all the particles within those regions, had them gained water vapor within those source regions or not. Thus, water does not necessarily come from those regions and they stop being 'source regions'.

*The reviewer is right. This is one of the limitations of the approach when analyzing contribution of remote sources. Particles can gain moisture in the regions placed between the defined moisture source and the target region, even in the target region. We have used extensively the same approach in many papers (see Gimeno et al, 2010 or Gimeno et al, 2013 as examples) and not always put in evidence this limitation in the text. However as our defined moisture regions were identified as the **major** moisture sources in the backward analysis (Vazquez et al, 2016) the contribution of the intermediate regions is much lower. We include figure 2 from Vazquez et al. 2006) that shows that intermediate regions are not net sources (particles reaching the Arctic region lost (not gained) moisture in these regions.*

*In any case we will include this limitation in the text of the revised version of the manuscript.*

[Figure]

**Figure 2.** (left column) Climatological annual and seasonal 10 day integrated $(E - P)$ values observed for the period 1979–2012, for all the particles bound for the Arctic domain, determined from backward tracking. Red (blue) colors represent moisture sources (sinks). Units are in mm d$^{-1}$. (middle column) Climatological annual and seasonal vertically integrated moisture flux values (vectors; measured in kg m$^{-1}$ s$^{-1}$) and respective divergence (shade; measured in mm d$^{-1}$). Data are from ERA Interim. (right column) Annual and seasonal moisture sources delimited only for those values of 10 day integrated $(E - P)$ greater than 0.4 mm/d. Each contour color represents one source: the dark blue line represents the Atlantic source, light blue the Siberian one, dark green the Pacific, light green the Gulf of Mexico, dark pink the Black Sea, light pink the Caspian Sea, light orange the Eastern Russia source, dark orange the Norwegian and Barents Seas, purple the China source, red the North American source, and garnet the Mediterranean source.

 In addition, not all the precipitated water comes from those 'sources', so, what happens with other particles that produce precipitation but were not within those 'source regions' ten days before precipitation?

*We don´t estimate precipitation in the Arctic but contribution of the major sources providing moisture for precipitation. Of course the rest of the particles are responsible for the rest of precipitation*

*We´ll include a comment on it in the revised version of the manuscript*

 Another question, are there enough 'particles' to properly characterize what happens with the smallest sub-regions defined in figure 1b (I am thinking in the results presented in figure 8)?

*The size of the target regions are bigger than many of the regions where the same methodology was used in previous studies. We calculated the average number of particle by source that reach daily the target regions (table below). The number is big enough.*

| | Central Arctic | CCA | Beaufort | Chuckchi | E. Siberian | Laptev | Kara | Barents | E. Greenland | Baffin Bay | Hudson Bay | St. Law | Bering Sea | Sea of Okhotsk |
|---|---|---|---|---|---|---|---|---|---|---|---|---|---|---|
| ATL | 4716 | 421, | 543, | 572, | 1102 | 1300 | 2733 | 9256 | 36900 | 40613 | 2961 | 17441 | 1829 | 4157 |
| PAC | 11059 | 4662 | 11023 | 10358 | 9275 | 2938 | 1698 | 3902 | 17773 | 32785 | 18675 | 15147 | 60348 | 33112 |
| NA | 21129 | 10426 | 9249 | 3415 | 3421 | 3510 | 4706 | 15336 | 69134 | 141067 | 59233 | 58688 | 6659 | 4794 |
| SIB | 22220 | 2829 | 9831 | 12078 | 21880 | 17030 | 16358 | 17235 | 7211 | 6184 | 4303 | 1090 | 41223 | 79120 |

*We´ll write a sentence in the revised paper to address this comment and the table will be included in the supplementary material*

Section 2.2.3 (should be 2.2.4): 'circulation types' are identified for four sections selected 'according to the positions of the major sources of moisture'. What are the sizes of those sections? Is there a minimum recommendable size? Is the method used to identify 'circulation types' sensible to the area selected? Are your results robust if you modify (nut much) those sections? Some of those sections share some common areas, does it affect to the latter interpretation of the circulation types? In addition, it would be advisable to identify those four sections in figure 1c.

*The size of the sections was 70 ºlatitude x 90ºlongitude. The analysis of changes in circulation types is complementary to the Lagrangian approach to check the coherence of the results. It is obvious that changes in the size of the sections can vary lightly the circulation types (Huth et al., 2008) but probably results of changes in the new types after/before the change point continue to be coherent with Lagrangian approach. We include in this comment a sample of this for the Atlantic section in fall by moving the domain 10º eastward and westward and by extending the domain 10º eastward (similar results, the patterns are very coherent)*

| CTC | As in the paper (60º) | Same size (60º) but moved 10º westward | Same size (60º) but moved 10º eastward | Extended 10º eastward |
|---|---|---|---|---|
| CTC1 |  |  |  |  |
| CTC2 |  |  |  |  |
| CTC3 |  |  |  |  |

CTC4

[Figure]

*In any case we have taken a domain higher than the used by Fettweiss et al. (2011) (the higher they used was 30ºx30º), who showed no significant differences in the circulation types for 4 different domain sizes. The use of a regional domain centered in the moisture source is justified to account for regional modes instead of annular ones which could not catch details in regional circulation.*

*We´ll include a discussion on this in the revised version and will change figure 1c as requested*

*Huth, R. et al. (2008): Classifications of Atmospheric Circulation Patterns – Recent Advances and Applications. Annals of the New York Academy of Sciences: Trends and Directions in Climate Research, 1146, p. 105–152*

Authors state that results in figure 6 suggest that 2003 is the most appropriate CP year. I do not find it so obvious. DS and MS series (figure 6) suggest that 2004 would be a better selection. Have authors tested if selecting one year or the other produces any difference? And, when writing about CPs, some explanation about change points identified using BinSeg and PELT should be provided. It is not clear in the text if more than one CP has been identified using those methods nor the implications that the existence of more than one CP in the SIE series would have in the interpretation of the results of this paper.

*We include the same analysis for changing 2003 by 2004. As you can see in the figure R2 (the equivalent to figure 7 in the text) results are quite similar. A comment on this will be included in the revised version of the manuscript*

[Figure]

*Figure R2 As Figure 7 in the manuscript but with 2004 as change point*

*In the description of figure 7 we commented the coincidence of change points found by AMOC (only one in the series) with any of the change points found by BinSeg and PELT (multiple change points). These two last approaches identify multiple change points (see the plots for the ADS series as an example of the series with more multiple changes or the plot for the 21 September DS series as an example of only one change in the three approaches, what is the most frequent case in DS and MS series). The idea of the paper is to identify the main change point to compare two periods (one with low ASI and the other with low). It is possible that any of the multiple subperiods identified by the other approaches merits analysis but it is out of the scope of this paper.*

*In any case we will include a comment in the revised version to suggest this for future work*

*AMOC*

[Figure]

*BinSeg*

[Figure]

*Pelt*

[Figure]

*AMOC*

[Figure]

*BinSeg*

[Figure]

*Pelt*

[Figure]

Some additional comments and typos: P7, paragraph describing figure 7: It is not explained anywhere that figure 7 includes the differences between mean values of MTP until 2003 and mean values after 2003 for every source region (this is my interpretation of what is represented in figure 7). The caption of figure 7 doesn't include this information either. Same comments can be applied to figure 8.

*We´ll do in the revised version of the manuscript*

I would suggest to re-plot figures 7 and 8 in order to include the information from the table in figure 7 and from table 1. It would be as easy as to plot with a thick (or filled) bar those differences that are statistically significant and with a thinner (empty) bar those differences that are not. In addition, plotting with a thicker line the horizontal bar indicating the 0 mm/day level would help to notice which sources increase/decrease their MTP contribution.

*We´ll do in the revised version of the manuscript for Figure 7, not possible for figure 8 (we keep the significativity table) because of the small size of the component figures*

Finally, no information about the statistical significance of the differences in total MTP is provided anywhere (again, it could be indicated by using a continuous or discontinuous line)

*Indicated with a red asterisk in the new figure 7*

[Figure]

**Figure 7. Differences between mean values of Moisture transport for precipitation (MTP) until 2003 and mean values after 2003 for every source region. Filled bars show those differences that are statistically significant at the 95% confidence level for decreases after the CP. Statistical significance of the differences in total MTP (sum of the four sources) is displayed with a red cross**

It would be easier to follow the comments in the text if Figure S1 and figure 9 included some labeled meridians (or at least, some longitudes in the outer area of the maps).

*We´ll do in the revised version of the manuscript*

TYPOS

*We´ll correct the typos and minor changes in the revised version of the manuscript*

Table S2: Are changes in the frequency of each class statistically significant?

*We have used a z-test to compare two sample proportions (Sprinthall, 2011). Statistical significant changes has been now included in Table S2 using asterisks*

- *Sprinthall, R. C. (2011). Basic Statistical Analysis (9th ed.). Pearson Education. ISBN 978-0-205-05217-2.*

---

## Author Comment (AC2) · 16 Mar 2018

**Anonymous Referee #2 "The perfect pattern of moisture transport for precipitation for Arctic sea ice melting"**

First, I find the title confusing and even misleading. Which perfect pattern is meant? Also, the statement "moisture transport for precipitation for Arctic sea ice melting" sounds rather awkward. I suggest to modify the title.

*The reviewer is right. The term "perfect pattern" is disconcerting. We´ll change the title to something less confusing such as "The pattern of long-term changes in the moisture transport for precipitation with Arctic sea ice melting". We prefer to keep the term moisture transport for precipitation because it is the usual term in previous studies using the same methodology*

The authors present a detailed analysis with clear graphical representations, however methodology description is unfortunately too vague to appreciate and understand the results. I invite the authors to explain the concept of the change points in relationship to its application to the Arctic sea ice extent data. Further, it is not clear how the results shown in Fig 6 have been obtained and how these results can be interpreted.

*We have used several methods to estimate change points in sea ice extension to detect when the main long-range change occurred. As usual in time series analysis a change point detection tries to identify times when the time series in mean or variance changed. In this case we were interested mainly in mean changes (in the Arctic sea extension the change means decrease, higher values before the change point and lower after it)*

*Three different change point detections were used for different sea ice extension time series, as explained in the manuscript. As an example we´ll included in the paper a new figure (above) for two series and one method, AMOC. The top plot represents the 13505-values Arctic ice extent anomalies series consisting of all days from 1st January 1980 to 31st December 2016. There are two horizontal lines in the left panel representing the mean of the values before and after the change point identified by the AMOC method (8660th day-22th September 2003). Those means are 0.27 and -0.91, respectively. The graphic at the bottom portrays the 444-values Arctic ice extent anomalies series consisting of all months from January 1980 to December 2016. As in the previous one, in the left panel there are two horizontal lines which correspond to the mean of the data before and after the AMOC change point (286-October 2003). Those means are 0.41 and -0.74, respectively.*

[Figure]

*Figure 6 tries to summarize results of the change points estimation in mean identified by the AMOC method for the four different kinds of series of Arctic ice extent anomalies from 1980 to 2016 (as described in the methods section). So:*

*\*A blue point refers to the change points in DS; for instance, the 21 July daily anomaly series (size of the series: 37 data points, representing 37 annual anomalies of the values of the sea ice extension for the 37 values on 21 July); occurred in 2004.*

*\* A red line portrays change points in the MS (July–October). For instance, for July monthly anomaly series (size of each series: 37 data points, representing 37 annual anomalies of the values of the sea ice extension for the 37 values in the average monthly July); occurred in 2004.*

*\* The single green square corresponds to the change point in the ADS, only one series of all daily anomalies (size of the series: 13,505 data points) built by ordering the daily anomalies in DS from 1 January 1980 to 31 December 2016 and the change point occurred the 22nd September 2003*

*\* The single purple line represents the change point in the AMS, only one series of all monthly anomalies (size of the series: 444 data points) built by ordering the monthly anomalies in MS from January 1980 to December 2016 and the change occurred in October 2003*

*This plot was designed to show the big coherence of results for the different built series, what permitted us to select 2003 as the CHANGE POINT of the sea ice extent for our analysis*

*We´ll try to explain a bit more the figure 6 with examples in the revised paper to do easier the reading and interpretation of the results*

Section 3.2 text is very descriptive and lacks interpretation. The methodology of the E-P analysis along the trajectories also needs to be better described. The moisture source regions are predefined from another study without any explanation - I invite the authors to explain the method in more details.

*In view to not enlarge the paper with wide methodological description (what has been done in many of our previous papers) we preferred to condense the information, lacking any details that probable difficult the correct understanding of the approach and the own meaning of the MTP, we´ll extend this description in the revised paper*

The wording itself "moisture transport for precipitation" sounds confusing and has to be rephrased and better defined.

*We prefer to keep the term moisture transport for precipitation because it is the usual term in previous studies using the same methodology, however we´ll try to explain better its meaning*

The sentence explaining the methodology ("Then, we selected all particles losing moisture, (e − p) < 0, at the sinks (whole Arctic or any of the subregions), and by adding e − p for all of these particles, we estimated moisture transport for precipitation from the source to the sink (E − P) < 0 at daily, monthly or yearly scales." ) needs more clarification with an extended explanation.

*We´ll extend the explanation in the revised paper*

This approach also doesn't imply that precipitation results exclusively from the moisture transport and local paper moisture re-circulation can also contribute.

*Of course, and this is without any doubt an important contribution, but not addressed in this study, limited to the influence of remote sources. We´ll add any sentence in the revised manuscript to account for this, which is important for the correct interpretation of the results*

I find a lot of similarities of this manuscript with another article by a coauthor of the present article Vasquez et al 2017 (www.mdpi.com/2073-4433/8/2/32/pdf), which is not somehow in the reference list. Can the authors put this manuscript in context of Vasquez et al 2017 explaining the novelty of the results?

*Although the objective of Vazquez et al. (2017) was too to analyze the effect of moisture transport on the Arctic ice melting and the Lagrangian approach is the same both studies differ a lot. In Vazquez et al. (2017) we analyzed the influence of the transport on the two most important sea ice minimum events (2007 and 2012) and the analysis is based mostly on an analysis of anomalies. In this paper we analyzed the long-term changes in the moisture transport concurrent with long-term changes in sea ice (sea ice decline). However we´ll add a comment on this in the introduction to contextualize the study.*

"We grouped individual days into four circulation types using the methodology developed by Fettweis et al. (2011 ) and explained in the Data and Methods sections." - in the Methods section the authors mention five (not four) circulation types and give no further explanations (which types, how they were defined...). Abbreviations used for the circulation types are not explained.

*The reviewer is right; there is an error in the number of classes. It will be corrected in the manuscript. More details on the circulation types methodology will be included in the revised version of the manuscript.*

*The abbreviations will be included in the methods section.*

What does it mean "the positive phase of the East Atlantic pattern" or " the negative phase of the East Atlantic/western Russia"?

*You are right. We have supposed readers familiar with teleconnection patterns and their phases. According to the NOAA definition "The term teleconnection pattern alludes to a to a recurring and persistent large-scale pattern of pressure and circulation anomalies that spans vast geographical areas. These patterns have strong influence on temperature, rainfall, storm tracks, and jet stream location/ intensity over vast areas and consequently are often assumed as responsible for abnormal weather patterns occurring simultaneously over seemingly vast distances". We included in the main text of the manuscript the known web page from NOAA http://www.cpc.ncep.noaa.gov/data/teledoc/telecontents.shtml where the patterns are described and their phases plotted. For instance, the East Atlantic pattern consists of a north-south dipole of anomaly centers spanning the North Atlantic from east to west with the positive phase with positive anomalies in the south pole, the one placed in the subtropics (see figure below)*

[Figure]

*We´ll describe a bit in the revised version of the manuscript what a teleconnection pattern is and the structure of those patterns referred in the main text*

I find that many statements in the Conclusions are not supported by the results. A major change seems to occur in 2003 - however unclear how this was obtained and what does it mean exactly (from the conclusions one can deduce that it means a drastic sea ice decline - I suppose the "change point" technique allows to detect that the mean SIE over the period after 2003 is

significantly smaller than before). And the "perfect pattern of MTP for Arctic sea ice melting consists of a general decrease in moisture transport in summer and an increase in fall and early winter", as stated in the Conclusions section, refers to this year as I understand (no longer mentioned in the Conclusions).

*We´ll re-write these sentences in the conclusions. You are right, the 2003 change means a drastic sea ice decline and a decrease in moisture transport in summer and an increase in fall and early winter after 2003 vs before 2003*

"This pattern is not only statistically significant but also consistent with Eulerian flux diagnosis, changes in circulation type frequency, and known mechanisms affecting snowfall or rainfall on ice in the Arctic." - which other known mechanisms affecting precipitation the authors refer to?

*Basically we are referring to the mechanisms summarized in the introduction*

*"Snowfall on sea ice enhances thermal insulation and thus reduces sea ice growth in winter (Leppäranta, 1993), but increases the surface albedo and thus reduces melt in spring and summer (Cheng et al., 2008). In contrast, rainfall is generally related to sea ice melt, and for both snowfall and rainfall, flooding over the ice favors the formation of superimposed ice and potentially increases in the Arctic sea ice thickness"*

*The implications of these mechanism coherent with our results would be:*

*A lower MTP in early summer (as occurred since 2003) is consistent with lower precipitation in snowfall decreasing the surface albedo and thus increasing melt (Cheng et al., 2008)*

*A lower MTP in late summer (as occurred since 2003) is consistent with less probability of occurrence of rainfall storms with possible flooding over the ice which would favor the formation of superimposed ice*

*A higher MTP in early fall (September) (as occurred since 2003) is consistent with higher precipitation as rainfall, something generally related to sea ice melt*

*A higher MTP in late fall and early winter (as occurred since 2003) is consistent with higher precipitation as snowfall, enhancing thermal insulation and thus reducing sea ice growth in (Leppäranta, 1993)*

*Of course to check these implications rigorously is clearly out of the scope of this manuscript, since it would imply to know details over the precipitation form (snow or rain) for the different Arctic regions with good temporal and geographical resolution, and even to analyze specific precipitation episodes to know if these are responsible for flooding or not.*

*The comment in the conclusion has as objective to reinforce the creditability of the significant results from the Lagrangian analysis.*

*We´ll extend the comment in the line of described in this comment*

"it is clear beyond doubt that an increment in moisture transport during this month favours ice melting, regardless of the source of moisture." - how is that clear? September's increase in MTP

according to the methodology used here (if I understood correctly) means increased local precipitation vs evaporation (not necessarily increased moisture transport), and its impact to the SIE has not been established in this study. There are previous studies showing that the linkage can be the other way that precipitation has increased because of the decreased sea ice extent (eg, Bintanja, R. & Selten, F. M. Future increases in Arctic precipitation linked to local evaporation and sea ice retreat. Nature 509, 479–482 (2014).).

*Not at all. I´m afraid that the reviewer has not understood properly the methodology and the meaning of MTP, probably for any lack of detail in the explanation of it (see previous comments). We´ll extend this explanation to avoid fails in the interpretation of the results. An increment in MTP in our study means exactly an increment in the moisture transported from the four major remote sources which then result in precipitation in the target region (Arctic region, subregions…). Changes in the precipitation in the Arctic could be of course due to changes in the moisture transport from remote sources but also and not less important to changes in evaporation from the own Arctic (a major factor, according to previous studies, but not evaluated in this paper).*

*We´ll try to clarify this in the revised version of the manuscript to avoid misunderstanding in the interpretation of the results*

"Snowfall is the dominant (almost unique) form of precipitation during most of the year, with the exception of late summer." - there is frequent rain during summer (and not only later summer), especially in the peripheral Arctic regions. Even in the central Arctic rain can occur in the very beginning of the melt period (like during SHEBA, eg Perovich et al 2002).

*We´ll re-write this to avoid so categorical affirmation*

" when precipitation is produced in the form of snowfall on sea ice, it enhances thermal insulation, and reduces sea ice growth in winter (Leppäranta, 1993), but increases the surface albedo, and thus reduces melt in spring and summer (Cheng et al., 2008). These phenomena justify the opposite change in moisture transport for fall and winter versus spring." - how can these phenomena justify any changes in moisture transport?

*These phenomena do not justify changes in the moisture transport but changes in the effect of precipitation on the sea ice. As we estimate changes in the moisture transport for precipitation (MTP), higher MTP results in more precipitation, this is the basis of the argument.*

The manuscript has to be checked for language and consistency - there are many vague, incomplete phrases.

*The first version was edited by a professional English service, in any case we´ll check carefully the revised version of the manuscript*

---

## Referee Report (RR1)

**2nd round revision of the manuscript submitted to ESD by Luis Gimeno-Sotelo, Raquel Nieto, Marta Vázquez, Luis Gimeno**

I thank the authors for addressing my previous major comments. The revised article text better explains the results and I recommend it for publication after taking into account some minor comments below.

**Title:**

Thank you for changing the title - however, I find the new title *"The pattern of long-term changes in the moisture transport for precipitation with Arctic sea ice melting"* still confusing. My suggestion would be something like this: "A new pattern of the moisture transport for precipitation related to the Arctic sea ice extent drastic decline" to better highlight the importance of the results in connection to the 2003 sea ice extent decline. But if the Editor agrees with the current title - I leave this to the authors to decide if they want to modify it.

**Abstract:**

1st sentence "In this study we use the term moisture transport for precipitation (MTP) for a target region as the moisture coming to this region from its major moisture sources that then results in precipitation over it." => "... resulting in precipitation over the target region". Please use the abbreviation MTP after introducing it.

"The pattern is not only statistically significant but also consistent with Eulerian fluxes diagnosis, changes in the frequency of circulation types" => I suggest changing to "The pattern is statistically significant and consistent with changes in the vertically integrated moisture fluxes and frequency of circulation types." It will make the abstract stronger if the authors add a sentence briefly specifying what are these consistent changes in the IVT and circulation types.

Please remove the statement "and any of the known mechanisms of the effects of the increments of precipitation as snowfall or rainfall on ice in the Arctic." - this sounds very vague, encompassing too many complex processes and feedbacks, and not the result of the present study.

**Page 2:**

The authors introduce two different abbreviations for the same term - "Arctic sea ice extension (SIE)" on page 2 and "Arctic sea ice (ASI) extension" on Page 4. I suggest to change "extension" to "extent" and use one abbreviation - Arctic SIE.

**Page 5:**
"defined in point 2.2.1." => section 2.2.1

"As in the previous one, there are two horizontal lines, which correspond to the mean of the data before and after the AMOC change point (the 286th month of the series, that is -

October 2003). Those means are 0.41 and -0.74, respectively."

=> please say specifically "to the SIE mean" instead of just "mean of the data"..

"Those means are 0.27 and - 0.91, respectively" and "those means are 0.41 and -0.74, respectively " - what are the units? I suppose million sq km if this is sea ice extent? Please check throughout the article text that the units are defined everywhere.

**Figure 2:**
1) Units are missing.. The yaxis title has to be more meaningful, i.e. "Sea ice extent monthly (daily) anomalies, units". That it is based on 37 years - this is additional information that has to be moved to the caption. The number of data points is redundant information (obvious from the plot).
2) xaxis: Please add a second y-axis on top indicating years; I suggest also changing "Day" -> "Days since..." and "Months since..."
3) Please make the red line thicker - together with years indicated on the top y-axis it should clearly show that it corresponds to 2003

**Comment on teleconnection patterns:**

*You are right. We have supposed readers familiar with teleconnection patterns and their phases. According to the NOAA definition "The term teleconnection pattern alludes to a to a recurring and persistent large-scale pattern of pressure and circulation anomalies that spans vast geographical areas. These patterns have strong influence on temperature, rainfall, storm tracks, and jet stream location/ intensity over vast areas and consequently are often assumed as responsible for abnormal weather patterns occurring simultaneously over seemingly vast distances". We included in the main text of the manuscript the known web page from NOAA http://www.cpc.ncep.noaa.gov/data/teledoc/telecontents.shtml where the patterns are described and their phases plotted. For instance, the East Atlantic pattern consists of a north- south dipole of anomaly centers spanning the North Atlantic from east to west with the positive phase with positive anomalies in the south pole, the one placed in the subtropics (see figure below)*

There is no need to explain what the term "teleconnection pattern" means - neither to me nor to the readers of ESD. I invited the authors to explain the specific patterns discussed in their study.

---

## Author Response (AR2)

REVIEWER 1

Most of my main concerns in the previous review have been answered to an acceptable degree.
As I said in my previous review, the analysis of CPs remains a little bit confusing. I do not understand why methods for the identification of multiple change points are important in the analysis presented in this paper and, thus, I would suggest removing the discussion about them in the final version of the text (any mention to results using BinSeg and PELT).

The use of three different CPs methods have two advantages, firstly as the three methods are different the coincidence of the CP detected using AMOC (only one by series) with any of the detected by the other two approaches (multiple CPs by series) reinforces the result reached by AMOC. Secondly, the existence of other CPs (as commented in the text) opens new possible studies on changes of MTP respect to any of the other CPs. So we prefer to maintain the analysis and discussion using the three CPs methods

Some of the corrections and new paragraphs included in the new version of the text are confusing:
The new title makes no sense. Do authors want to emphasize the existence of a pattern in the 'changes in the moisture transport...', or the existence of changes in 'the pattern of moisture transport for precipitation ASSOCIATED with Arctic sea ice melting'? I guess it is the second one. Could the new title be something like: 'Changes in the long-term pattern of moisture transport for precipitation associated with Arctic sea ice melting'

We have changed again the title. The new one is" A new pattern of the moisture transport for precipitation related to the Arctic sea ice extent drastic decline" as suggested by the reviewer 2

Abstract, same question as before: 'We have identified the patterns of change in...' or 'We have identified changes in the patterns of...'? Changed
The new version of the first paragraph in section 2.2.2 is very confusing: review

Changed, now is written so:

"We have used several methods to estimate change points in Arctic sea ice (ASI) extension. As usual in time series analysis a change point detection tries to identify times when the time series in mean or variance changed, in this case we were interested mainly in changes in mean. Changes in mean for the Arctic sea ice extension is equivalent to a decrease, higher ASI extension values before the change point and lower after it".

The paragraph beginning in page 9, line 4 is confusing and should be reviewed...

Changed. We have removed the term "is consistent" by implies and we have written the most frequent type of precipitation by season, snowfall, rainfall... into brackets. We think that now it is less confusing

Conclusions (page 12 line 4): Same question as in the title: 'The pattern of change' or 'a change of the pattern'? I find it is the pattern of MTP what changes. It is difficult to talk about a pattern of change from a single change point and thus, a single change case.

Changed

Some suggestions:
P5L1: the three different change point DETECTION methods Done
P6L2: ...the defined as the major moisture sources... -> ...those defined as major moisture sources... Done
P6L8: ...including the particles coming from the own Arctic region -> ...including the particles coming from the Arctic region itself Done
P6L10: ...2005), THIS works ... -> ...2005), it works... Done
P6L13: ...big enough -> ...high enough Done
P8L32: (red crosses) -> (red asterisks) Done
P12L3: ...which then results in precipitation over the Arctic (MTP) -> ...which then results in changes in the precipitation over the Arctic Done

2nd round revision of the manuscript submitted to ESD by Luis Gimeno-Sotelo, Raquel Nieto, Marta Vázquez, Luis Gimeno I thank the authors for addressing my previous major comments. The revised article text better explains the results and I recommend it for publication after taking into account some minor comments below.

Title: Thank you for changing the title - however, I find the new title "The pattern of long-term changes in the moisture transport for precipitation with Arctic sea ice melting" still confusing. My suggestion would be something like this: "A new pattern of the moisture transport for precipitation related to the Arctic sea ice extent drastic decline" to better highlight the importance of the results in connection to the 2003 sea ice extent decline. But if the Editor agrees with the current title - I leave this to the authors to decide if they want to modify it.

We have changed again the title. The new one is" A new pattern of the moisture transport for precipitation related to the Arctic sea ice extent drastic decline" as suggested by the reviewer

Abstract:

1st sentence "In this study we use the term moisture transport for precipitation (MTP) for a target region as the moisture coming to this region from its major moisture sources that then results in precipitation over it." => "... resulting in precipitation over the target region". Please use the abbreviation MTP after introducing it.

Changed

"The pattern is not only statistically significant but also consistent with Eulerian fluxes diagnosis, changes in the frequency of circulation types" => I suggest changing to "The pattern is statistically significant and consistent with changes in the vertically integrated moisture fluxes and frequency of circulation types." It will make the abstract stronger if the authors add a sentence briefly specifying what are these consistent changes in the IVT and circulation types.

Changed

Please remove the statement "and any of the known mechanisms of the effects of the increments of precipitation as snowfall or rainfall on ice in the Arctic." - this sounds very vague, encompassing too many complex processes and feedbacks, and not the result of the present study.

Removed

Page 2:

The authors introduce two different abbreviations for the same term - "Arctic sea ice extension (SIE)" on page 2 and "Arctic sea ice (ASI) extension" on Page 4. I suggest to change "extension" to "extent" and use one abbreviation - Arctic SIE.

Done. We have changed ASI extension to Arctic SIE and extension to extent

Page 5:

"defined in point 2.2.1." => section 2.2.1 "As in the previous one, there are two horizontal lines, which correspond to the mean of the data before and after the AMOC change point (the 286th month of the series, that is - October 2003). Those means are 0.41 and -0.74, respectively."

=> please say specifically "to the SIE mean" instead of just "mean of the data"..

Changed

"Those means are 0.27 and - 0.91, respectively" and "those means are 0.41 and -0.74, respectively " - what are the units? I suppose million sq km if this is sea ice extent? Please check throughout the article text that the units are defined everywhere.

You are right. Units is in million sq km. Added

Figure 2:

1) Units are missing.. The yaxis title has to be more meaningful, i.e. "Sea ice extent monthly (daily) anomalies, units". That it is based on 37 years - this is additional information that has to be moved to the caption. The number of data points is redundant information (obvious from the plot).

2) xaxis: Please add a second y-axis on top indicating years; I suggest also changing "Day" -> "Days since..." and "Months since..."

3) Please make the red line thicker - together with years indicated on the top y-axis it should clearly show that it corresponds to 2003

Done

Comment on teleconnection patterns:

There is no need to explain what the term "teleconnection pattern" means - neither to me nor to the readers of ESD. I invited the authors to explain the specific patterns discussed in their study.

Removed